# Spatiotemporal orchestration of calcium-cAMP oscillations on AKAP/AC nanodomains is governed by an incoherent feedforward loop

**Lingxia Qiao**[1,2]**, Michael Getz**[3]**, Ben Gross**[2]**, Brian Tenner**[4]**, Jin Zhang**[1,5,6]***,
**Padmini Rangamani**[1,2]*

**1** Department of Pharmacology, University of California San Diego, San Diego, California, United States of America, **2** Department of Mechanical and Aerospace Engineering, University of California San Diego, San Diego, California, United States of America, **3** Luddy School of Informatics, Computing, and Engineering, Indiana University, Bloomington, Indiana, United States of America, **4** SomaLogic, San Diego, California, United States of America, **5** Department of Bioengineering, University of California San Diego, San Diego, California, United States of America, **6** Department of Chemistry and Biochemistry, University of California San Diego, San Diego, California, United States of America

* jzhang32@health.ucsd.edu (JZ); prangamani@health.ucsd.edu (PR)

**Data Availability Statement:** All the code to generate data and figures is available from: https://

## Abstract

The nanoscale organization of enzymes associated with the dynamics of second messengers is critical for ensuring compartmentation and localization of signaling molecules in cells. Specifically, the spatiotemporal orchestration of cAMP and $Ca^{2+}$ oscillations is critical for many cellular functions. Previous experimental studies have shown that the formation of nanodomains of A-kinase anchoring protein 79/150 (AKAP150) and adenylyl cyclase 8 (AC8) on the surface of pancreatic MIN6 β cells modulates the phase of $Ca^{2+}$-cAMP oscillations from out-of-phase to in-phase. In this work, we develop computational models of the $Ca^{2+}$/cAMP pathway and AKAP/AC nanodomain formation that give rise to the two important predictions: instead of an arbitrary phase difference, the out-of-phase $Ca^{2+}$/cAMP oscillation reaches $Ca^{2+}$ trough and cAMP peak simultaneously, which is defined as inversely out-of-phase; the in-phase and inversely out-of-phase oscillations associated with $Ca^{2+}$-cAMP dynamics on and away from the nanodomains can be explained by an incoherent feedforward loop. Factors such as cellular surface-to-volume ratio, compartment size, and distance between nanodomains do not affect the existence of in-phase or inversely out-of-phase $Ca^{2+}$/cAMP oscillation, but cellular surface-to-volume ratio and compartment size can affect the time delay for the inversely out-of-phase $Ca^{2+}$/cAMP oscillation while the distance between two nanodomains does not. Finally, we predict that both the Turing pattern-generated nanodomains and experimentally measured nanodomains demonstrate the existence of in-phase and inversely out-of-phase $Ca^{2+}$/cAMP oscillation when the AC8 is at a low level, consistent with the behavior of an incoherent feedforward loop. These findings unveil the key circuit motif that governs cAMP and $Ca^{2+}$ oscillations and advance our understanding of how nanodomains can lead to spatial compartmentation of second messengers.

github.com/RangamaniLabUCSD/Qiao_et_al_
cAMP-Calcium-on-AKAP-AC-nanodomains.

**Funding:** This work was supported by the National Institute of Health Grant R01 DK073368 (to J.Z.), the Air Force Office of Scientific Research (AFOSR) Multidisciplinary University Research Initiative (MURI) Grant FA9550-18-1-0051 (to P.R.), and Army Research Office W911NF2310249 to (P.R.). The funders had no role in study design, data collection and analysis, decision to publish, or preparation of the manuscript.

**Competing interests:** The authors have declared that no competing interests exist.

## Author summary

Cyclic adenosine monophosphate (cAMP) and $Ca^{2+}$ are key molecules that relay signals from the cell membrane to downstream molecules. The temporal and spatial distribution of cAMP and $Ca^{2+}$ within the cell can be regulated by clusters formed by specific molecules. Previous studies have shown that nanodomains of A-kinase anchoring protein 79/150 (AKAP150) and adenylyl cyclase 8 (AC8) on the surface of pancreatic MIN6 β cells modulates $Ca^{2+}$-cAMP oscillations from non-zero to zero phase difference, i.e., from out-of-phase to in-phase. By developing a computational model of the $Ca^{2+}$/cAMP pathway, we found that the non-zero phase difference is not an arbitrary value: it makes $Ca^{2+}$ reach trough and cAMP reach peak simultaneously. We defined this out-of-phase behavior as inversely out-of-phase, and revealed that this behavior and in-phase behavior can be explained by an incoherent feedforward loop. Biophysical properties of cells and nanodomain distributions do not affect the existence of in-phase or inversely out-of-phase behaviors, but may affect the time delay for the inversely out-of-phase behavior. Furthermore, AKAP/AC nanodomains from a Turing pattern or experiments show consistent simulation results with the incoherent feedforward loop. This study improves our understanding of underlying mechanisms of $Ca^{2+}$/cAMP oscillation, shedding light on controlling $Ca^{2+}$ and cAMP through nanodomains.

## 1 Introduction

Cyclic adenosine monophosphate (cAMP) and $Ca^{2+}$ are key second messengers that regulate several cellular functions, including muscle contraction, neuronal excitability, cell migration, metabolism, endocytosis, plasma membrane repair, and immune function [1–6]. The execution of these cellular functions relies on both the temporal dynamics and the spatial distribution of cAMP and $Ca^{2+}$. The temporal dynamics of cAMP and $Ca^{2+}$ play crucial roles in many biological systems. For example, changes in the cAMP level and downstream protein kinase A (PKA) are connected with tumorigenesis, invasion, metastasis and drug resistance [7–9], and an increase in cAMP levels inhibits hepatocellular carcinoma cell proliferation [10] and induces the apoptosis of glioblastoma cells [11]; $Ca^{2+}$ frequency can be decoded by downstream $Ca^{2+}$ sensors and translated into distinct cellular responses [12–16]. In addition to tightly regulated temporal control, cAMP and $Ca^{2+}$ are highly spatially compartmentalized at the plasma membrane and cellular organelles [17–19]. If this spatial compartmentalization is disrupted, it may lead to changes in signaling pathways and gene expression [20–22], as well as pathological conditions such as cell apoptosis [22], abnormal insulin secretion [23], heart dysfunction [3, 24, 25], familial breast cancer [26], and schizophrenia [27].

The temporal dynamics of cAMP and $Ca^{2+}$ usually demonstrate oscillatory behaviors in many cell types including neurons [28, 29], cardiomyocytes [30], and pancreatic β cells [31]. Given the importance of spatiotemporal orchestration of $Ca^{2+}$ and cAMP function, understanding the biochemical and biophysical mechanisms that govern their oscillations is very important. Mathematical and computational modeling can aid in deciphering how different cellular features play a role in governing these oscillations [32–37]. Previously, we showed that the frequency of $Ca^{2+}$-cAMP oscillations depends on the geometry of dendritic spines in neurons [38, 39]. In this study, we focus on cAMP and $Ca^{2+}$ oscillations in pancreatic β cells because of their critical role in insulin secretion. The pathway regulating the insulin secretion in pancreatic β cells is as follows: an increase in blood glucose leads to the increase in the

cytosolic ATP/ADP ratio in the pancreatic β cell [40]; then, ATP-sensitive potassium channel ($K_{ATP}$) channel closes, inducing the influx of $Ca^{2+}$ and subsequent insulin vesicle exocytosis [41, 42]. During this insulin secretion process, the cAMP pathway is also activated: increased $Ca^{2+}$ level also stimulates the activation of adenylyl cyclase type 8 (AC8), and AC8 improves the cAMP production, followed by protein kinase A (PKA) activation. Furthermore, cAMP level is also controlled by the $Ca^{2+}$-dependent PDE1C (phosphodiesterase 1C; denoted as PDE for simplicity) in pancreatic β cells, because this PDE is activated by $Ca^{2+}$ and then mediates the transient cAMP decreases [32]. The activated cAMP/PKA pathway affects cellular excitability, $Ca^{2+}$ signals, exocytosis, cell viability, and cell–cell interactions [31, 43, 44]. Disturbances in this pathway may be associated with diabetes [45].

The cross-talk between $Ca^{2+}$ and cAMP is vital for the proper function of pancreatic β cells. Such cross-talk is achieved by the highly connected spatiotemporal organization between cAMP and $Ca^{2+}$ in pancreatic β cells. The temporal dynamics of $Ca^{2+}$ and cAMP in pancreatic β cells show synchronized oscillations, and this oscillation is initiated and modulated by PKA [46]. Recently, experiments showed that the synchronized oscillation of these two second messengers exhibits distinct phase delay at different membrane locations. For example, the membrane nanodomain formed by plasma membrane (PM)-localized scaffold protein A-kinase anchoring protein 79 (AKAP79; rodent ortholog AKAP150) [32] (Fig 1A). AKAP79/150 is a multivalent scaffold protein that associates with itself to form nanodomains. Additionally, AKAP79/150s also can recruit signaling molecules to cell membrane, including PKA, the voltage-gated $Ca^{2+}$ channel $Ca_V1.2$, Protein Kinase C (PKC), the $Ca^{2+}$/calmodulin-dependent protein phosphatase calcineurin, $Ca^{2+}$-sensitive ACs, and AMPA receptors [47]. On membrane nanodomains formed by A-kinase anchoring proteins 79/150 (AKAP79/150) and AC8 (referred to as the AKAP/AC nanodomains), $Ca^{2+}$ and cAMP were found to oscillate in-phase, i.e., without any phase difference. Away from the membrane AKAP/AC nanodomains, $Ca^{2+}$ and cAMP oscillate out-of-phase. This discovery that membrane nanodomains of AKAP79/150 and AC8 can modulate the in- and out-of-phase oscillations of cAMP-$Ca^{2+}$ leads to interesting questions about the biochemical and biophysical mechanisms that regulate the formation of these nanodomains and coordinate the spatiotemporal dynamics of these critical second messengers. Can the dynamics of in- and out-of-phase oscillations of $Ca^{2+}$ and cAMP be explained by a simple network motif? Does subcellular compartment size affect these phase shifts? How do patterns of nanodomains form on the membrane and how do multiple nanodomains affect these oscillatory patterns? We sought to answer these questions using computational modeling.

## 2 Results

In this work, we developed two classes of models—a 3D reaction-diffusion model to simulate the dynamics of cAMP and $Ca^{2+}$ and a Turing pattern model with steric hindrace to simulate the formation of AKAP/AC nanodomains. For the 3D reaction-diffusion model, we defined preexisting AKAP/AC nanodomains and focused on the dynamics of cAMP and $Ca^{2+}$. We used this model to investigate how activities of AC and PDE, cellular compartment size, and pattern of nanodomains affect the in- and out-of-phase oscillations of $Ca^{2+}$ and cAMP (Fig 1B). To understand how nanodomains might form, we used a Turing model with steric repulsion [48] to characterize the regimes in which stable AKAP/AC nanodomains might arise. Finally, to investigate how the nanodomain patterns might affect the in- and out-of-phase oscillations, we used patterns from both the Turing model and experimentally generated STORM data as the initial distribution of AKAP/AC nanodomains and predicted that both the pattern of AKAP/AC nanodomains and AC activities are critical in controlling the phase

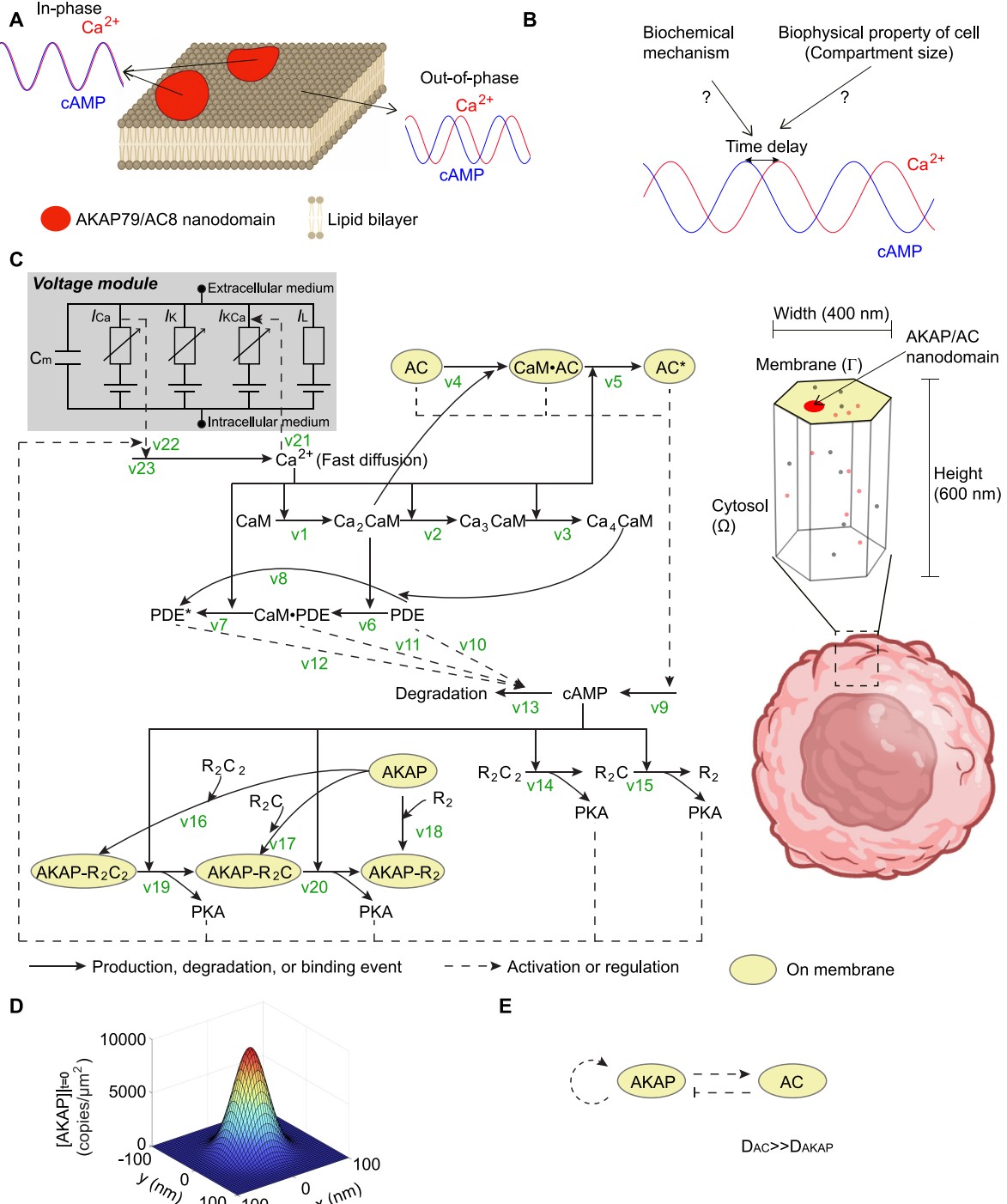

**Fig 1. The phase regulation of Ca²⁺-cAMP oscillation and corresponding mathematical modeling.** (A) Schematic showing out-of-phase Ca²⁺-cAMP oscillation outside the nanodomain and in-phase behavior when localized to AKAP/AC nanodomains. This schematic is designed based on experiments in [32] (created with BioRender.com). (B) In this work, we aim to explore the biochemical mechanism of the phase difference (or time delay) between Ca²⁺ and cAMP oscillation and study the possible contributing factors. (C) Mathematical modeling of the AKAP-Ca²⁺-cAMP circuit. On the left-hand side, the diagram of the signaling pathway is shown; the solid arrow indicates production, degradation, or binding events, and the dashed arrow indicates the regulation effect that usually does not consume the reactants. The voltage module (highlighted in gray) includes a capacitor with the membrane capacitance $C_m$ and four ion channels: Ca²⁺ channel, K⁺ channel, Ca²⁺ gated K⁺ channel, and leak channel. Currents for ion channels are represented by $I_{Ca}$, $I_K$, $I_{KCa}$, and $I_L$, where the subscript indicates the specific ion channel. On the right-hand side, the simulation domain is a hexagonal prism, which is only a small compartment of one cell (created with BioRender.com). In this compartment, the top surface (yellow area) denotes the cell membrane; the

AKAP/AC nanodomain (the large patch in red) is located on the cell membrane; the volume under the top surface is cytosol. Molecules (dots in gray and orange) can diffuse in the cytosol or on the membrane depending on the location of the molecule. (D) A single AKAP/AC nanodomain was modeled using a Gaussian distribution. (E) Interactions between AC and AKAP that can generate a Turing pattern.

behavior of $Ca^{2+}$-cAMP. These results provide insights into the biochemical and biophysical mechanisms that regulate the spatiotemporal dynamics of $Ca^{2+}$ and cAMP.

## 2.1 Phase oscillations of calcium and cAMP are driven by active AC and PDE

We first simulated the dynamics of $Ca^{2+}$ and cAMP using the 3D reaction-diffusion model (see Reaction-diffusion model subsection in Methods for details). We briefly describe the reactions below. We considered 22 state variables in total (Fig 1C and S1 Table), where 13 chemical species are in the cytosol (non-highlighted species in Fig 1C) and the rest are on the membrane (highlighted in yellow in Fig 1C). The reactions among these chemical species are briefly introduced below (S2 and S3 Tables): the increase of $Ca^{2+}$ in the cytosol leads to the activation of inactive PDE (Cyclic nucleotide phosphodiesterase) in the cytosol (v6–8) and inactive AC (Adenylyl cyclase) on AKAP/AC nanodomains through CaM (calmodulin) (v1–3 and v4–5); next, the active AC (denoted as AC$^*$) on AKAP/AC nanodomains enhances the synthesis of cAMP (v9), and the active PDE (denoted as PDE$^*$) improves the degradation of the cAMP in the cytosol (v10–13); the change in the cAMP level will affect PKA (protein kinase A) in the cytosol (v14–20), and the latter will regulate the dynamics of $Ca^{2+}$ by interacting with the voltage module. In the voltage module (gray box in Fig 1C), the membrane is regarded as a capacitor with the capacitance $C_m$, and four ion channels are considered: $Ca^{2+}$ channel, $K^+$ channel, $Ca^{2+}$ gated $K^+$ channel, and leak channel. The currents for these four ion channels are represented by $I_{Ca}$, $I_K$, $I_{KCa}$, and $I_L$. The current that flows through the membrane is equal to the negative of the sum of ion channel currents, i.e., $C_m \frac{dV}{dt} = -(I_{Ca} + I_K + I_{KCa} + I_L)$, where $V$ denotes the membrane voltage. The $Ca^{2+}$ affects the $Ca^{2+}$ gated $K^+$ channel (v21), and the $Ca^{2+}$ channel controls the $Ca^{2+}$ influx (v22).

Kinetic parameters for the above reactions are from [32], which were listed in S4 Table. Among these parameters, those for binding/unbinding events between $Ca^{2+}$ and CaM are from [49], and those for the membrane voltage model come from [46]. Other kinetic parameters in [32] were estimated empirically (labeled as "[Estimated]" in S4 Table) or obtained by fitting to the experimental data (labeled as "[Constraint]" in S4 Table). The experimental data used include not only the ranges of the molecule concentration that the sensor can monitor but also the relative changes of biosensor fluorescence for $Ca^{2+}$ and cAMP in [32]. The diffusion coefficients and initial values were estimated empirically to ensure that their values are biologically plausible. The settings for AKAP/AC nanodomain and compartment are the same as the calculations in [32] (see Reaction-diffusion model subsection in Methods for details): at time 0 there is only one AKAP/AC nanodomain located at the center of the cell membrane, which is modeled by a Gaussian distribution with standard deviation of 25 nm (Fig 1D); the compartment has a height of 600 nm and width of 400 nm (Fig 2A). Furthermore, we chose periodic boundaries for the six surfaces surrounding the compartment, so that many compartments can assemble to form a large cell domain without any discontinuity. These kinetic parameters ensure that values of $Ca^{2+}$ concentration, cAMP concentration, and membrane voltage are biologically plausible (panel A in S1 Fig). Specifically, the $Ca^{2+}$ concentration simulated by the model is between 0 and 1.5 μM (panel A(ii) in S1 Fig), which is within the same scale of the $Ca^{2+}$ concentration in internal $Ca^{2+}$ store (0.1–0.5 μM from [50]). The cAMP

concentration in the model ranges from 0 to 0.2 μM (panel A(ii) in S1 Fig), which falls within the detectable range for sensors ([32]). The membrane voltage in the model also shows the same scale as that in [51] (panel A(iii) in S1 Fig). The cAMP and PKA levels far from the nano-domain are lower than those on the nanodomain (panels A(ii) and (vi) in S1 Fig), forming a spatial heterogeneity that might be crucial for proper cellular functions [52].

Next, we used the above model to simulate the dynamics of $Ca^{2+}$ and cAMP when there is only one AKAP/AC nanodomain existing in the center of the membrane. Due to the limited experimental data, which monitored intracellular cAMP only at the cell membrane rather than in the cytosol [32], we also only focused on the dynamics of $Ca^{2+}$ and cAMP at the cell membrane. Because the AKAP/AC nanodomain is radially symmetric, we only need to focus on dynamics in the line from the center to the right edge of the cell membrane (white line in Fig 2B) and used $x$ to denote the distance from the center to the location of interest at cell membrane. We found that, the time delay between $Ca^{2+}$ and cAMP peak time is zero when $x = 0$ and near 25 seconds when $x$ is larger than 50 nm (Fig 2B), indicating the in-phase behavior at the AKAP/AC nanodomain and out-of-phase behavior outside AKAP/AC nanodomain. This conclusion is consistent with the simulation results in [32]. The transition happens at $x = 48$ nm, which is nearly two times of standard deviation of the Gaussian distribution; this suggests that the out-of-phase behavior also occurs at a low but non-zero level of AC*.

To explore why there is a sharp transition of the time delay when changing from AKAP/AC nanodomain to the general membrane, we plotted the dynamics of $Ca^{2+}$ and cAMP for different values of $x$ (Fig 2C). The dynamics of $Ca^{2+}$ remain unchanged at different locations (red lines in Fig 2C) because of the fast diffusion of $Ca^{2+}$. In contrast, the dynamics of cAMP are different at different locations: its level decreases when $x$ increases (Fig 2C). Furthermore, cAMP has two peaks for intermediate $x$, but the relative height varies (Fig 2C): the peak that is in-phase with $Ca^{2+}$ is higher than the out-of-phase peak at small $x$, that is, at the center of the nanodomain, but is smaller compared with the out-of-phase peak at large $x$, i.e., far from the nanodomain. Here, the out-of-phase peak denotes any non-zero difference in the phase between two oscillatory trajectories. We also found that the in-phase peaks at different locations likely occur at the same time, and so do out-of-phase peaks (dashed gray line in Fig 2C). These results suggest that the sharp transition of the time delay is caused by the swapping between the in-phase and out-of-phase peaks. To better visualize the delay of peaks between $Ca^{2+}$ and cAMP, we plotted the normalized dynamics of $Ca^{2+}$ and cAMP (Fig 2D). Here, nor-malization indicates that the concentration is divided by the maximum value at a specific loca-tion during the time interval [0, 1200] seconds. It can be seen that peaks of cAMP at $x = 0$ are in-phase with $Ca^{2+}$ at $x = 0$, those at $x = 49$ nm are either in-phase or out-of-phase with $Ca^{2+}$, and those at $x = 200$ nm are out-of-phase with $Ca^{2+}$. These results further support that the in-phase peak dominates on the AKAP/AC nanodomain while the out-of-phase peak dominates outside the AKAP/AC nanodomain.

To understand the kinetics of in-phase and out-of-phase cAMP peaks, we first analyzed the dynamics of the immediate upstream species: $Ca^{2+}$, active AC (AC*), and active PDE (PDE*). We plotted the dynamics of concentrations of these three species at the cell membrane (Fig 2E), and then calculated the trough and peak time (marked by the plus sign or triangle). Here, the trough (or peak) time is defined as the time when the concentration of species reaches the minimum (or maximum) value. For $Ca^{2+}$, due to its fast diffusion, the trough and peak times remain unchanged when the location $x$ varies (upper panel in Fig 2E). As for AC*, its peak and trough times also do not change when the location $x$ varies between 0 and 20 nm, but these peak and trough times cannot be defined at $x > 20$ nm because of the low AC* level outside the AKAP/AC nanodomain (middle panel in Fig 2E). The PDE* shows similar behavior as $Ca^{2+}$, that is, the trough and peak times for PDE* are fixed no matter how the location $x$ changes

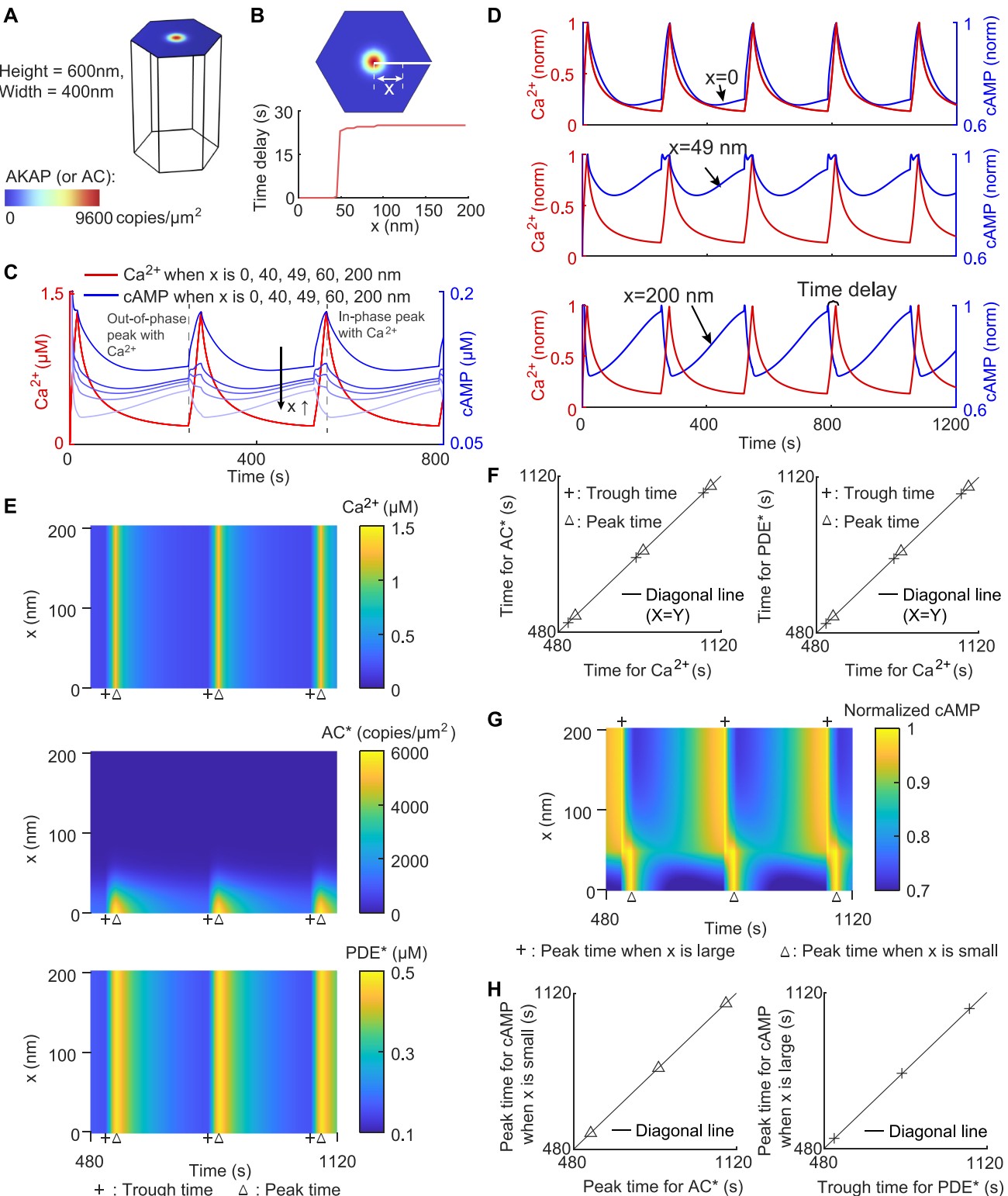

**Fig 2. Phase oscillations of Ca$^{2+}$ and cAMP are driven by active AC and PDE.** (A) The simulation domain and initial condition of AC and AKAP. The compartment and the assumption of one AKAP/AC nanodomain are the same as those in [32]. (B) The time delay between Ca$^{2+}$ and cAMP as a function of the distance $x$ to the AKAP/AC nanodomain. The time delay is defined as the difference of peak time between Ca$^{2+}$ and cAMP. (C) The non-normalized dynamics of Ca$^{2+}$ and cAMP at $x = 0$, 40, 49, 60, 200 nm. The vertical dashed lines label the time when cAMP achieves the peak. The blue color indicates the value of $x$: the lighter the blue is, the larger the $x$ is. (D) The normalized dynamics of Ca$^{2+}$ (in red) and cAMP (in blue) at $x = 0$ nm

(upper panel), $x$ = 49 nm (middle panel), and $x$ = 200 nm (lower panel). (E) Kymographs depicting the dynamics for Ca$^{2+}$, active AC (AC*), and active PDE (PDE*) at different locations at the cell membrane. The x coordinate is the time, and the y coordinate is the distance $x$ present in (B). Trough time and peak are indicated by a plus sign and triangle, respectively. Here, the peak time is the time when the concentration of species reaches the maximal value, and the trough time the minimal value. (F) Comparisons between Ca$^{2+}$ trough time and AC* trough time (left; plus sign), between Ca$^{2+}$ peak time and AC* peak time (left; triangle), between Ca$^{2+}$ trough time and PDE* trough time (right; plus sign), and between Ca$^{2+}$ peak time and PDE* peak time (right; triangle). The diagonal line indicates the equality of the x-axis and y-axis. (G) Kymograph depicting cAMP dynamics. The color intensity indicates the normalized cAMP level. The plus sign and triangle denote the cAMP peak time when x is large and small, respectively. (H) Comparisons between AC* peak time and cAMP peak time on the AKAP/AC nanodomain (left), and between PDE* trough time and cAMP peak time outside the AKAP/AC nanodomain (right).

(lower panel in Fig 2E). Next, we compared these times between Ca$^{2+}$ and AC* and between Ca$^{2+}$ and PDE* (Fig 2F). We found that, the AC* trough time at $x < 20$ nm is the same as that for Ca$^{2+}$, and so is the peak time (left panel in Fig 2F). Similarly, the trough and peak times for PDE* also match the trough and peak times for Ca$^{2+}$, respectively (right panel in Fig 2F). These consistencies in the trough and peak time may be caused by the strong activation from Ca$^{2+}$ to AC* (v4–5 in S2 Table) and from Ca$^{2+}$ to PDE* (v1–8 in S2 Table). Furthermore, these consistencies indicate that the AC* on the nanodomain and PDE* at the entire membrane exhibit in-phase oscillation with Ca$^{2+}$.

After analyzing the dynamics of upstream species of cAMP, we turned to explore how these dynamics may affect in-phase and out-of-phase behaviors of cAMP. We plotted the dynamics of the normalized cAMP level at different locations at the cell membrane (Fig 2G), and then compared the cAMP peak time with PDE* trough time or AC* peak time (Fig 2H). The cAMP concentration for each $x$ is normalized by the maximal value during the time interval [0, 1200] seconds. From the dynamics of the normalized cAMP level (Fig 2G), we can find that the cAMP peak time shifts from left to right when the location $x$ decreases. Moreover, the cAMP peak time is the same as the AC* peak time when $x$ is small (left panel in Fig 2H), but equal to the PDE* trough time when $x$ is large (right panel in Fig 2H). These results suggest the in-phase AC*-cAMP oscillation on the AKAP/AC nanodomain and inversely out-of-phase PDE*-cAMP oscillation outside the AKAP/AC nanodomain, where the "inversely" is defined as one oscillator reaching the peak while the other reaches the trough. Since AC* catalyzes the cAMP synthesis (v9 in S2 Table) and PDE* enhances the cAMP degradation (v12 in S2 Table), these inversely out-of-phase and in-phase behaviors may result from the competition between AC*-induced cAMP production and PDE*-induced cAMP degradation. On the AKAP/AC nanodomain, the AC* level is much higher than PDE*, and thus the activation from AC* dominates, leading to in-phase AC*-cAMP oscillation. In contrast, outside the AKAP/AC nanodomain, the AC* shows low values, and thus the cAMP levels are mainly governed by the action of PDE*; PDE* inhibits the cAMP level by enhancing the cAMP degradation, resulting in inversely out-of-phase PDE*-cAMP oscillation. When combined with the observation of in-phase Ca$^{2+}$-AC* oscillation on the nanodomain and in-phase Ca$^{2+}$-PDE* oscillation at the entire membrane, the cAMP is in-phase with Ca$^{2+}$ on the AKAP/AC nanodomain but inversely out-of-phase with Ca$^{2+}$ outside the AKAP/AC nanodomain.

## 2.2 Experimental data supports model predictions of in- and inversely out-of-phase calcium-cAMP oscillations

We summarized the mechanism of in-phase and inversely out-of-phase Ca$^{2+}$-cAMP oscillations in Fig 3A. The origin of the stimulus is the Ca$^{2+}$ oscillation, which is uniformly distributed spatially due to rapid diffusion. Ca$^{2+}$ dynamics affects the phase of the cAMP by two species AC* and PDE*, whose dynamics are as follows: the PDE* oscillation at the entire cell membrane is in-phase with Ca$^{2+}$ through the activation from Ca$^{2+}$ to PDE* (v1–3 and v6–8 in

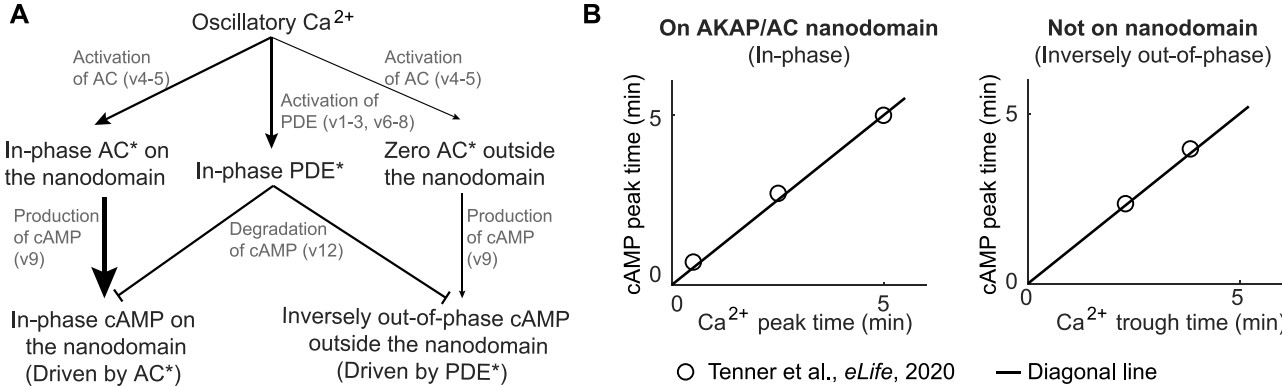

**Fig 3. Experimental data supports model predictions of in- and inversely out-of-phase Ca²⁺-cAMP oscillations.** (A) Schematic of the dependence of time delay between Ca²⁺ and cAMP on the interplay between active AC and PDE. The arrow thickness indicates the regulation strength: the thicker the arrow is, the stronger the regulation is. On the AKAP/AC nanodomain, the active AC dominates, driving the in-phase Ca²⁺-cAMP oscillation. However, the active PDE dominates outside the AKAP/AC nanodomain, leading to inversely out-of-phase Ca²⁺-cAMP oscillation. (B) The experimentally observed in-phase Ca²⁺-cAMP oscillation on the AKAP/AC nanodomain and inversely out-of-phase Ca²⁺-cAMP oscillation outside the AKAP/AC nanodomain. The in-phase Ca²⁺-cAMP oscillation is illustrated by the same cAMP peak time and Ca²⁺ peak time; the inversely out-of-phase Ca²⁺-cAMP oscillation is indicated by the same cAMP peak time and Ca²⁺ trough time. These peak or trough time data (circular makers) are from Tenner et al., *eLife*, 2020 [32], and the black line denotes the diagonal line.

S2 Table). On the AKAP/AC nanodomain, AC* is not only in high concentration but also in-phase with Ca²⁺ since the Ca²⁺ catalyzes the activation of AC* (v4–5 in S2 Table) whereas outside the AKAP/AC nanodomain, AC* is in low concentration, because AC is in low concentration outside the AKAP/AC nanodomain. As a result, on the AKAP/AC nanodomain, the enriched AC* enhances the cAMP production (v9 in S2 Table), driving the cAMP oscillation become in-phase with AC*; outside the nanodomain, the AC* level is much smaller than the PDE* level, and the latter degrades cAMP (v12 in S2 Table), causing inversely out-of-phase cAMP oscillation with PDE*. Since AC* on the nanodomain and PDE* on the entire membrane are both in-phase with Ca²⁺, the cAMP oscillation is also in-phase with Ca²⁺ on the nanodomain and inversely out-of-phase with Ca²⁺ outside the nanodomain. To further verify the role of the AC* and PDE* in the phase difference, we also simulated the dynamics of Ca²⁺ and cAMP under the two following conditions: the first is to delete the AKAP/AC nanodomain, and the second is to delete both AKAP/AC nanodomain and PDE*. For the first case, we found that the in-phase behavior disappears but the out-of-phase behavior exists on the entire cell membrane (panel B in S1 Fig). Note that no AKAP/AC nanodomain means a zero level of AC*. Thus, the simulation result in the first case suggests that the AC* contributes to the in-phase behavior instead of the out-of-phase behavior, which is consistent with the above mechanism. For the second case, we set the PDE* level to zero in the absence of AKAP/AC nanodomain, and Ca²⁺ still oscillates but cAMP cannot (S2 Fig), which is consistent with Figure 5E in [53] where cells were treated with the PDE inhibitor 3-isobutyl-1-methylxanthine (IBMX). Thus, this simulation result in the second case indicates that PDE* contributes to the cAMP oscillation. Taken together, AC* and PDE* are crucial for the oscillation and phase difference.

We tested the model predictions against previously published experimental data in [32]. First, we extracted the time when the response of the cAMP biosensor reaches the highest value, and then denoted this time as the cAMP peak time. In [32], two types of cAMP biosensors were used. The AKAP/AC nanodomain-specific cAMP biosensor provides the cAMP peak time on the AKAP/AC nanodomain, and the plasma membrane-specific cAMP biosensor gives the cAMP peak time outside the AKAP/AC nanodomain. These peak times from two

types of cAMP biosensors are the y-coordinates in Fig 3B. Next, we extracted the time when the response of the $Ca^{2+}$ biosensor reaches the highest value and lowest value, and denoted them as $Ca^{2+}$ peak and $Ca^{2+}$ trough time respectively (x coordinates in Fig 3B). We found that the cAMP peak time on the AKAP/AC nanodomain is close to $Ca^{2+}$ peak time (left panel in Fig 3B), suggesting the in-phase $Ca^{2+}$-cAMP oscillation. However, the cAMP peak time outside the AKAP/AC nanodomain is the same as the $Ca^{2+}$ trough time (right panel in Fig 3B). This means that the phase difference between the $Ca^{2+}$ and cAMP is exactly half of the period, which is inversely out-of-phase. Therefore, these experiment results are consistent with the numerical predictions from our model with respect to the in-phase and inversely out-of-phase $Ca^{2+}$-cAMP oscillation. It should be noted that, although the in-phase and out-of-phase behaviors have been extensively studied in [32], the property that $Ca^{2+}$ trough and cAMP peak occurs simultaneously in the out-of-phase behavior was first discovered in this work.

## 2.3 A simple incoherent feedforward loop explains in-phase oscillation and inversely out-of-phase oscillation

The above analyses demonstrate the role of AC* and PDE* in regulating the phase difference between $Ca^{2+}$ and cAMP. However, the complexity of the biochemical circuit involved in the regulation of $Ca^{2+}$ and cAMP dynamics (Fig 1C) prevents us from identifying if AC* and PDE* are sufficient conditions for such phase regulation. We next investigated if a simple circuit model could shed light on the main control elements of the phase difference. The signal, which mimics $Ca^{2+}$ concentration, is set to be a sine wave plus a constant 1.1, where the constant 1.1 ensures the positive sign of the signal (Fig 4A). Furthermore, we assumed that the signal affects output paradoxically. On one hand, the signal improves the production of output and on the other hand, the signal increases the output degradation (Fig 4B). The circuit with two pathways with opposite effects from the input to the output is usually called as incoherent feedforward loop [54], and can exhibit different biological functions under certain conditions [54–61]. These regulatory reactions are used to mimic the effect of $Ca^{2+}$ on cAMP caused by AC* and PDE*: $Ca^{2+}$ enhances the cAMP production by the activation of AC; at the same time, $Ca^{2+}$ improves the cAMP degradation by the activation of PDE. Therefore, the output can be regarded as the cAMP. The mathematical model of this simple incoherent feedforward loop is written as follows:

$$\frac{dx}{dt} = vS(t) + k_1 - m[S(t) + k_2]x \tag{1}$$

where $S(t)$ and $x$ denote the concentration of signal and output, respectively. $v$ is the activation strength caused by signal, $k_1$ is the basal production rate constant, $m$ is the degradation rate constant and $k_2$ captures the feedback of $x$ on the degradation. Without loss of generality, we can set $m = 1$ and obtain (see Simple ODE model subsection in Methods for details):

$$\frac{dx}{dt} = vS(t) + k_1 - [S(t) + k_2]x. \tag{2}$$

The solution of this first-order linear ODE when $t$ goes to infinity can be written as follows (see Simple ODE model subsection in Methods for details):

$$x(t) = \left(v - \frac{k_1}{k_2}\right)e^{-I(s)-k_2t}\int_0^t S(s)e^{I(s)+k_2s}ds + \frac{k_1}{k_2} \triangleq \left(v - \frac{k_1}{k_2}\right)F(t) + \frac{k_1}{k_2}, \tag{3}$$

where $I(s)$ is $\int_0^s S(z)dz$. $F(t)$ is defined as $e^{-I(t)-k_2t}\int_0^t S(s)e^{I(s)+k_2s}ds$ for simplicity.

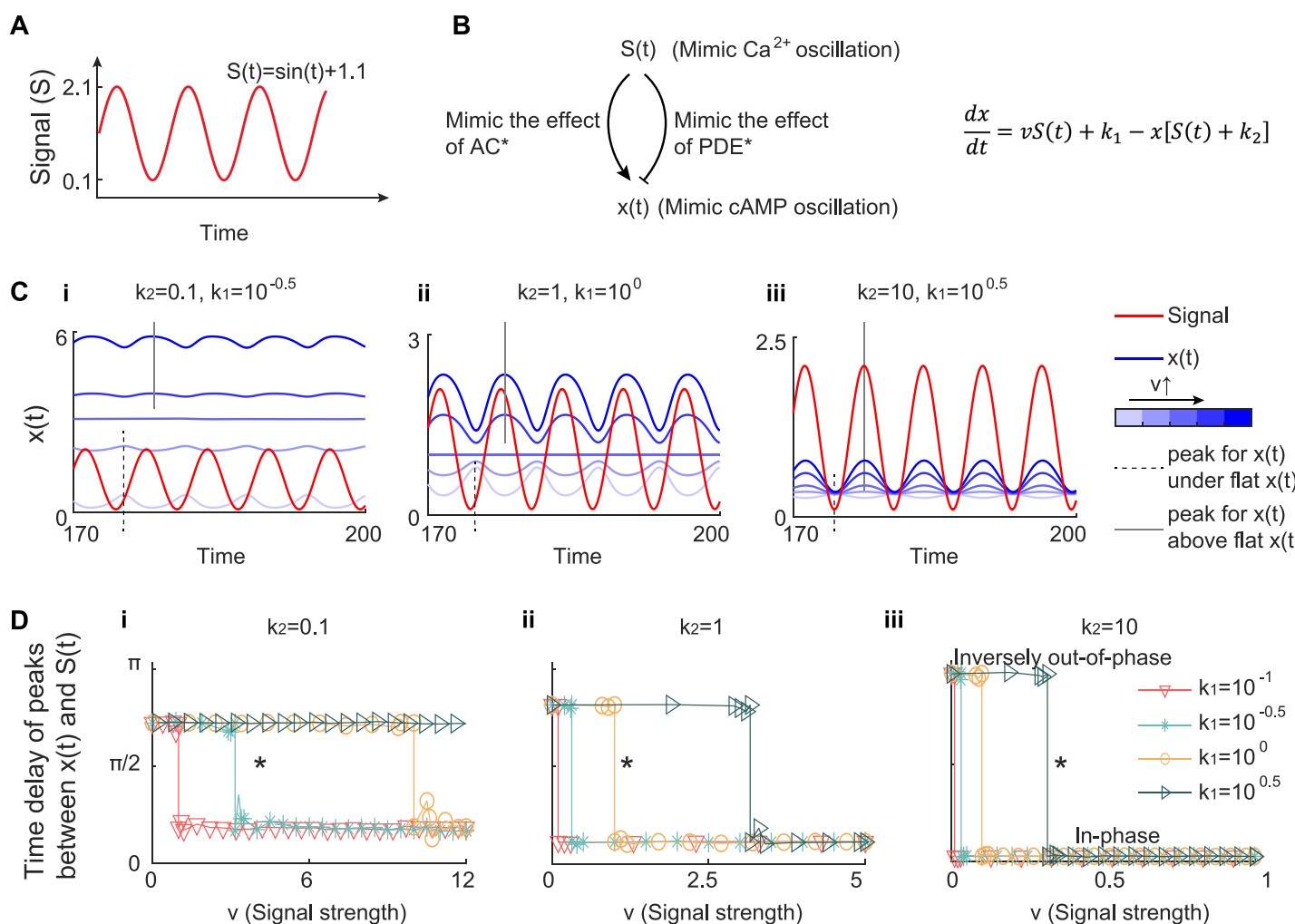

**Fig 4. A simple incoherent feedforward loop explains in-phase oscillation and inversely out-of-phase oscillation.** (A) The signal used in the simple ODE model. We used a sine wave $sin(t) + 1.1$ as the signal $S(t)$ to mimic the dynamics of $Ca^{2+}$, where 1.1 is to ensure the positive sign of the signal. (B) Schematic of the ODE model. We constructed a simple circuit with only two regulatory links: one is the activation from the signal $S(t)$ to the output $x(t)$; the other is the inhibition from the signal $S(t)$ to the output $x(t)$. This circuit captures the positive role of $Ca^{2+}$ not only in the cAMP production through active AC but also in the cAMP degradation through active PDE. The equation describing the dynamics of $x(t)$ is shown in the right panel. (C) The dynamics of $S(t)$ (red) and $x(t)$ (blue) under different values of $k_1$ and $k_2$, and $v$. Values of $k_1$ and $k_2$ are labeled over the plots. Values of $v$ are indicated by the intensity of blue color (the darker the blue is, the higher the $v$ is): $v = [0, 2, 3.1, 4, 6]$ (i), $v = [0, 0.5, 1, 2, 3]$ (ii), and $v = [0, 0.5, 1, 2, 3]$ (iii). The trough and peak of $x(t)$ are marked by the dashed and solid lines, respectively. (D) The time delay between $S(t)$ and $x(t)$ as a function of the signal strength $v$ for different values of $k_1$ and $k_2$. Three values of $k_2$ are considered: 0.1 (i), 1 (ii), and 10 (iii). In each panel, $k_1$ is changed from $10^{-1}$ to $10^{0.5}$, shown in different colors and markers. The stars in each panel indicate the parameters used in (C). The time delay has been proven to have only two values when the activation strength $v$ is increased while keeping other kinetic parameters fixed (see Simple ODE model subsection in Methods for details), and thus the fluctuations in the plot of time delay versus $v$ come from numerical errors.

The integral in the Eq (3) lacks a closed form solution. Therefore, we turned to numerical solutions to evaluate how the different parameters affected the output as a function of the input. We varied the value of the activation strength $v$, which mimics the effect of AC*, from small (away from the nanodomain) to large (center of the nanodomain). We postulated that if the oscillation behavior of $x(t)$ is inversely out-of-phase with small $v$ and then becomes in-phase with increased $v$, then this simple incoherent feedforward loop is sufficient to generate similar results observed in the 3D reaction-diffusion model. We found that, with small values of the basal production rate constant $k_1$ and the $x$-induced degradation rate $k_2$, increasing

activation strength $v$ cannot lead to the transition from inversely out-of-phase to in-phase behavior between the input and output (Fig 4C(i) and 4C(ii)), but given large $k_1$ and $k_2$ we can recover the proposed $v$ dependence (Fig 4C(iii)). In Fig 4C(iii), when $v$ is small, the output peak time is the same as the signal trough time (the dashed line in Fig 4C(iii)), indicating the inversely out-of-phase oscillation; when $v$ is large, the output peak time is the signal peak time (the solid line in Fig 4C(iii)), suggesting an in-phase oscillation. While in-phase oscillation and inversely out-of-phase oscillation are reproduced by the simple ODE model, the transition between these two behaviors is not exactly the same as the full model in Fig 2A. As explained previously, the sharp transition in the full model is caused by the swapping between in-phase and out-of-phase peaks, but in the simple ODE model we didn't observe the emergence of two peaks. The failure to observe two peaks in the simple ODE model may be because this simple ODE model neglects other reactions that also consume cAMP, such as the binding of cAMP and $R_2C_2$ (v14 in Fig 1C), the binding of cAMP and AKAP–$R_2C_2$ (v19 in Fig 1C).

To quantitatively identify the phase difference, we calculated the time delay of peaks between $x(t)$ and $S(t)$ for several combinations of $k_1$ and $k_2$ (Fig 4D). The relationship between the time delay and phase behavior is as follows—if the time delay is half of the signal period, that is, $\pi$, the inversely out-of-phase oscillation is achieved while zero time delay corresponds to the in-phase oscillation. We found that the $x$-induced degradation rate $k_2$ is more important than the basal production rate constant $k_1$ in producing the desired $v$ dependence. If $k_2$ is small, it is impossible to achieve inversely out-of-phase and in-phase oscillations even with different values of $k_1$ (Fig 4D(i) and 4D(ii)); only the large $k_2$ can lead to inversely out-of-phase and in-phase oscillations under different values of $v$ (Fig 4D(iii)). Furthermore, the role of the basal production rate constant $k_1$ is to shift the critical value of activation strength $v$ that causes the transition from the inversely out-of-phase to the in-phase. These numerical results are also validated by analyzing the Eq (3). Since $x'(t) = \left(v - \frac{k_1}{k_2}\right)F'(t)$, the time when $x(t) = 0$ is the same as that when $F'(t) = 0$. Therefore, the peak time of $x(t)$ can only be the peak time or the trough time of $F(t)$, that is, only two values. This is consistent with the numerical simulations that there are only two values of time delay in Fig 4D with varied $v$. Moreover, the peak time of $x(t)$ is the trough time of $F(t)$ when $v < \frac{k_1}{k_2}$ and is the peak time of $F(t)$ when $v > \frac{k_1}{k_2}$ (see Simple ODE model subsection in Methods for details), corresponding to the time delay before and after transition in Fig 4D, respectively. Especially, when $k_2$ is large enough, $F(t)$ has the same peak and trough time as the signal $S(t)$ (see Simple ODE model subsection in Methods for details), and thus the peak time of $x(t)$ is the trough time of the signal $S(t)$ when $v < \frac{k_1}{k_2}$ and the peak time of $S(t)$ when $v > \frac{k_1}{k_2}$, consistent with inversely out-of-phase and in-phase behaviors observed in Fig 4D(iii). In conclusion, the exploration of the simple incoherent feedforward loop not only validates the sufficiency of the role of AC* and PDE* in driving phase difference, but also suggests the mechanism of inversely out-of-phase and in-phase oscillations—a large basal degradation rate (i.e., $k_2$ in the Eq (2)). Thus, by simulating the dynamics of the output governed by the Eq (2), we validated that such a simple incoherent feedforward loop is sufficient to produce in-phase and inversely out-of-phase oscillations.

## 2.4 Simulation results predict the role of AC in driving the in-phase behavior and decreasing time delay for inversely out-of-phase behavior

Since the above analyses for a simple incoherent feedforward loop suggest the role of AC* in controlling the phase behavior of $Ca^{2+}$-cAMP, we tested whether the AC activity in the 3D reaction-diffusion model also plays a similar role. Therefore, we decreased the maximal concentration of AC to 75%, 50%, and 25% of the original value, and then simulated the 3D

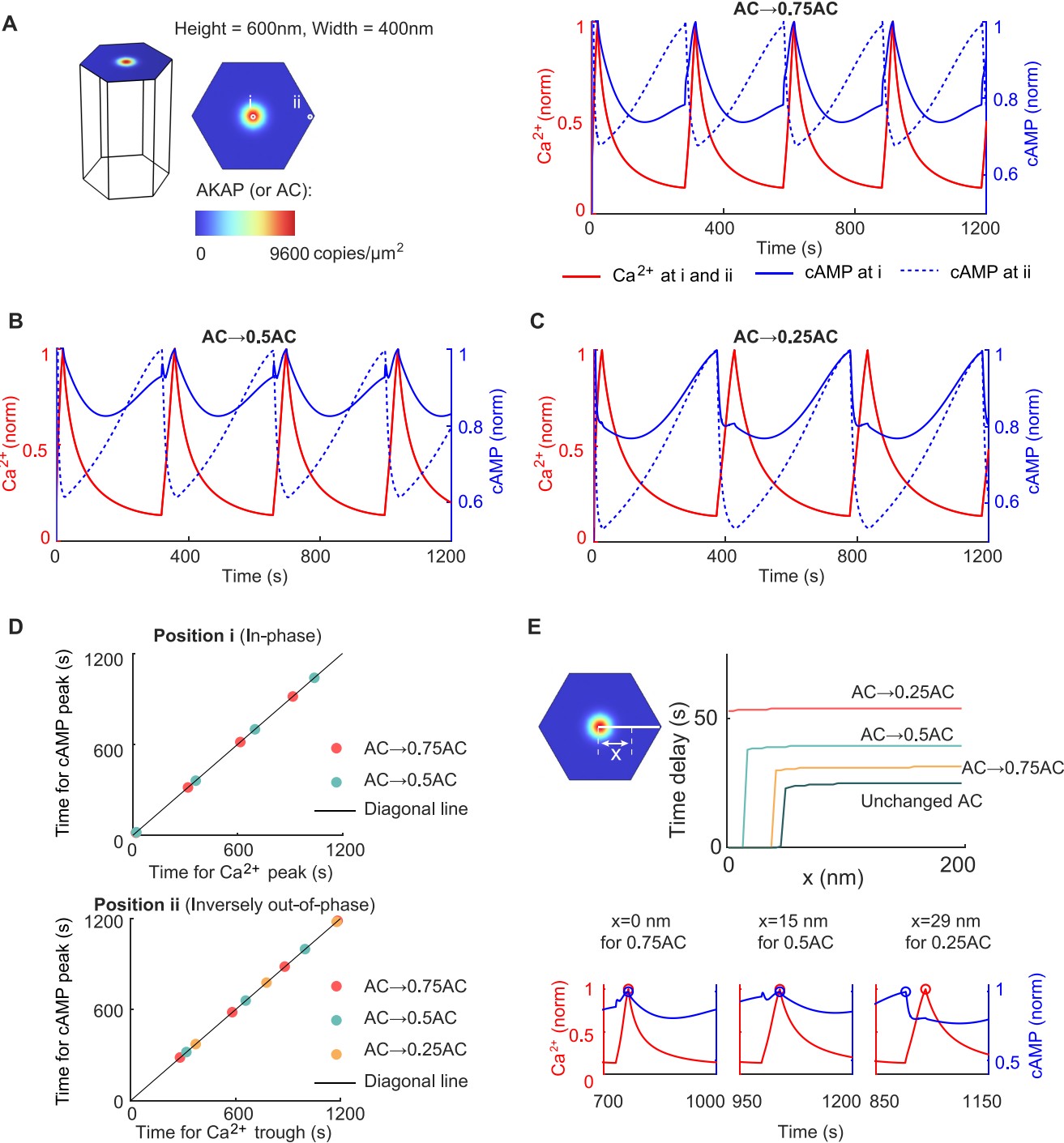

**Fig 5. Simulation results predict the role of AC in driving the in-phase behavior and decreasing time delay for inversely out-of-phase behavior.** (A-C) The dynamics of Ca$^{2+}$ and cAMP at different locations when the AC level is varied. The AC concentration is 75% (A), 50% (B), and 25% (C) of the original value. (D) The in-phase and inversely out-of-phase cAMP behavior in (A-C). (E) The time delay of peaks between Ca$^{2+}$ and cAMP as a function of the distance $x$ in (A-B). The $x$ has the same definition as that in Fig 2B.

reaction-diffusion model (Fig 5A–5C). We found that the in-phase behavior on AKAP/AC nanodomains and inversely out-of-phase behavior outside AKAP/AC nanodomains still exist for the 75% and 50% cases (Fig 5A, 5B and 5D), but only inversely out-of-phase behavior appears for the 25% case (Fig 5C and 5D). These results are consistent with the effect of activation strength $v$, because a small value of $v$ usually leads to out-of-phase behavior. In addition to the effect of AC activity on the phase difference between $Ca^{2+}$ and cAMP, AC activity also influences the time delay for the inversely out-of-phase behavior, that is, the higher the AC activity is, the shorter the time delay is (Fig 5E).

## 2.5 Cellular compartment size determines the time delay for the inversely out-of-phase calcium-cAMP oscillation

Having analyzed the biochemical mechanisms that regulate the $Ca^{2+}$-cAMP phase difference, we next turn our attention to the biophysical properties that may regulate the spatiotemporal dynamics of cell signaling, including the compartment size to investigate how it influences this phase difference. Since the cell is a 3D structure, the height and width of the compartment reflect the cell size to some degree: the higher or wider the compartment, the larger the probability that a cell will have a large size. Researchers have identified the cell shape as an important factor in regulating cellular signaling [62–64], and thus we expect that the compartment size also affects the oscillation behavior. First, we tested the effect of the compartment size by doubling the height (Fig 6A), doubling the width (Fig 6B), or doubling both the height and width (Fig 6C). This effectively changes the surface-to-volume ratio of the cellular compartment. Similar to the original compartment (Fig 2A), these large compartments all lead to in-phase cAMP oscillation on the AKAP/AC nanodomains and inversely out-of-phase cAMP oscillation outside the AKAP/AC nanodomains (Fig 6A–6D). However, the time delay for the inversely out-of-phase behavior differs (Fig 6E): it is near 25 seconds for the original compartment (black dashed line in Fig 6E), 50 seconds for the compartments with the double width or double height (red and cyan lines in Fig 6E), and 120 seconds for the compartment with double width and height (yellow line in Fig 6E). The compartment with the double width and that with double height have different sizes but similar time delay, indicating that the compartment size cannot monotonically determine the time delay (also see S3 Fig). Nevertheless, by comparing these large compartments with the original compartment, we predict that the time delay for the inversely out-of-phase cAMP oscillation depends on the compartment size, and the large compartment size tends to induce a long time delay. Thus, the cell with a large volume tends to exhibit a long time delay. Although this prediction is not based on a whole-cell simulation, it fits the intuition that the large cell may take a long time to transform the signal as the molecule needs a long time to diffuse to specific locations.

## 2.6 Prediction of the time delay from idealized Turing pattern-based AKAP/AC nanodomains

Thus far, we focused on the impact of preexisting nanodomains. We next investigated the conditions that could lead to the formation of nanodomains. Due to the clustered distributions of AKAP and AC molecules ([32] and S4 Fig), we considered the patterning model to describe the AKAP/AC nanodomain formation. Although AC molecules do not form a periodic pattern according to the diverse distance between two nearest nanodomains and non-uniformed nanodomain size (S4 Fig), we still chose the Turing model. The reasons for this model choice are as follows: 1) the significant immobile fraction (near 42.3%) of AC molecules (Figure 3—figure supplement 3 in [32]) is consistent with the long-lived property of Turing patterns; 2) the STORM data for AKAP and AC in different cell types is limited [65, 66], and thus the

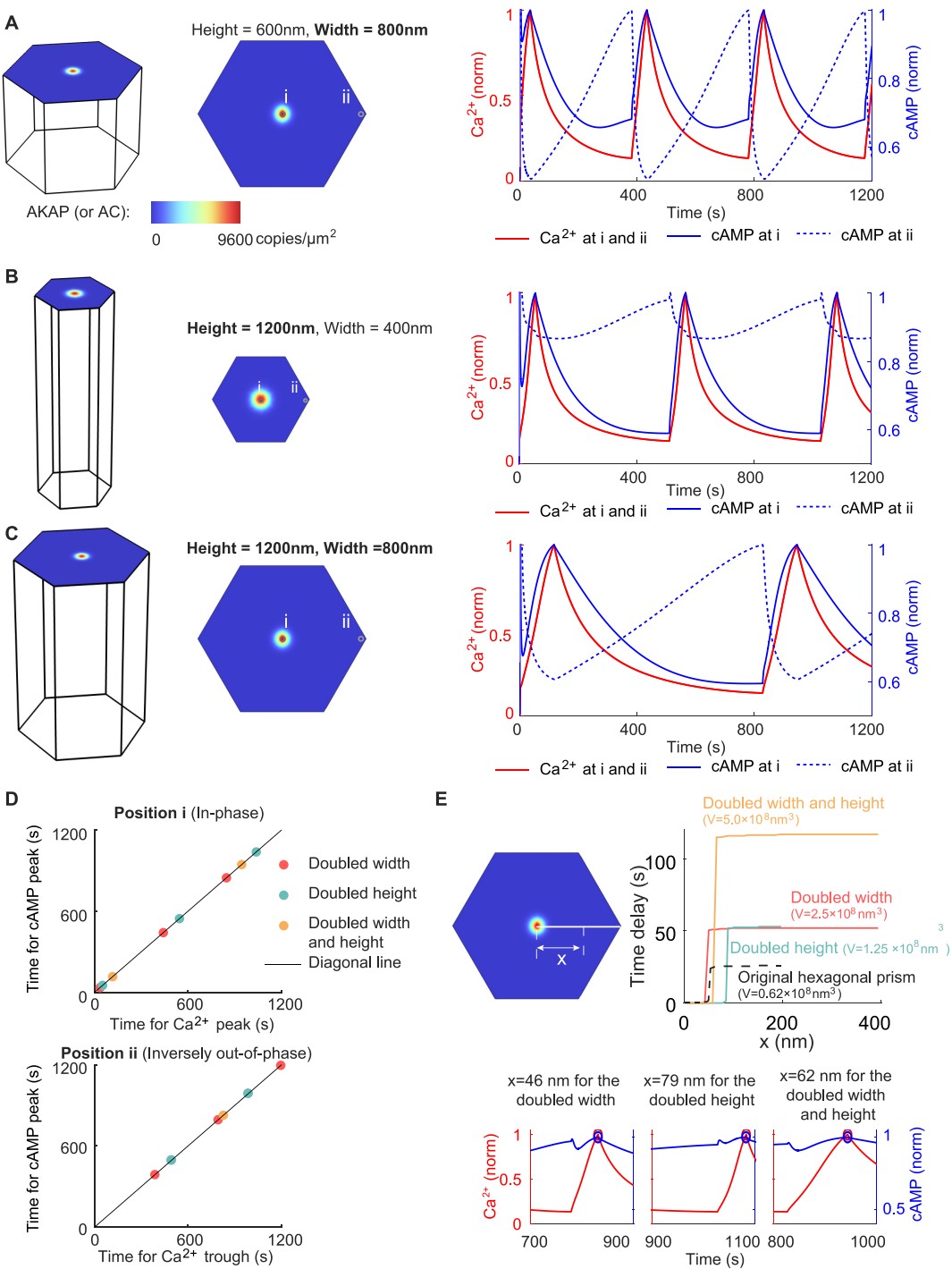

**Fig 6. Cellular compartment size determines the time delay for the inversely out-of-phase Ca²⁺-cAMP oscillation.** (A) The $Ca^{2+}$ and cAMP dynamics for a double-width compartment. Dynamics on the AKAP/AC nanodomain (i) and at the edge of the compartment (ii) are shown. (B-C) Same plots as (A) except the compartment size. The compartment is doubled in height in (B) and doubled in both height and width in (C). (D) The in-phase and inversely out-of-phase oscillations in (A-C). (E) The time delay in (A-C) as a function of the distance $x$ to the nanodomain. The lower panel shows the magnified view of the $Ca^{2+}$ and cAMP dynamics when $x$ is close to the critical value, which is defined as the time when the transition of time delay occurs.

identification of the pattern formed by AC and AKAP lacks solid evidence. Therefore, we chose the Turing model for mathematical tractability.

Inspired by work done by others on the formation of membrane receptor nanodomains [48, 67–70], we focused on a Turing model with steric interactions [48]. This model is summarized as follows (Fig 1E):

$$
\begin{cases}
\dfrac{\partial r}{\partial t} = -b\left(r - \dfrac{s}{\bar{s}}\dfrac{1-r-s}{1-\bar{r}-\bar{s}}\bar{r}\right) - m_1\dfrac{1-r-s}{1-\bar{r}-\bar{s}}(r-\bar{r}) + m_2\dfrac{1-r-s}{1-\bar{r}-\bar{s}}r\bar{r}(s-\bar{s}) \\
\qquad\quad + D_r\nabla[(1-s)\nabla r + r\nabla s] \\[4pt]
\dfrac{\partial s}{\partial t} = -\beta\left(s - \dfrac{1-r-s}{1-\bar{r}-\bar{s}}\bar{s}\right) + \mu\dfrac{1-r-s}{1-\bar{r}-\bar{s}}s\bar{s}(s-\bar{s}) + D_s\nabla[(1-r)\nabla s + s\nabla r].
\end{cases}
\tag{4}
$$

In the Eq (4), $r$ and $s$ denote the normalized AC and AKAP concentration, respectively. AKAP works as a scaffold to recruit AC to cell membrane, and AC inhibits the recruitment of AKAP by steric repulsion similar to the model presented in [48]. $\bar{r}$ and $\bar{s}$ are the value of $r$ and $s$ at homogeneous steady state. $b$ and $\beta$ are the unbinding rates of the AC and AKAP from the cell membrane, respectively; $m_1$ and $m_2$ are the binding rates of the AC to cell membrane caused by itself and the AKAP, respectively; $\mu$ is the self-recruitment rate of AKAP; $D_r$ and $D_s$ are diffusion coefficients. The values of these kinetic parameters are listed in S7 Table. However, the assumption that AKAP/AC nanodomains are formed by the Turing model is a hypothesis, where future experiments are required to validate the existence of the above biochemical reactions. The values of these kinetic parameters may also change with the dynamics of nanodomains.

We simulated the above Turing model in a hexagonal domain of 800 nm for different combinations of kinetic parameters (Fig 7A). We found that three patterns arise: a homogeneous steady state where AC (denoted by $r$) and AKAP (denoted by $s$) are both uniformly distributed (lower left panel in Fig 7A), Turing pattern 1 where AC and AKAP are co-localized (lower middle panel in Fig 7A), and Turing pattern 2 where AC and AKAP repel each other (lower right panel in Fig 7A). We found that the Turing pattern usually occurs when the ratio of the diffusion coefficient of AC to the diffusion coefficient of AKAP (i.e., $D_r/D_s$) is large (area in red and yellow in the upper left panel in Fig 7A), which is consistent with the condition of Turing instability. However, the large ratio leads to the transition from Turing pattern 1 to Turing pattern 2. Taken together with the fact that AC is co-localized with AKAP in MIN6 β cells [32], our model predicts that the ratio of diffusion coefficients $D_r/D_s$ is a critical determinant of nanodomain formation with Turing pattern 1. We note that while Turing pattern 2 is a mathematically admissible solution, its relevance to cell signaling is as yet unknown. Furthermore, colocalization and nanodomain formation are not sensitive to the unbinding rate of the AC (denoted by $b$) but have a strict requirement on the unbinding rate of the AKAP, binding rate of the AC to cell membrane caused by AKAP, and the self-recruitment rate of AKAP (denoted by $\beta$, $m_2$, and $\mu$, respectively) (upper middle and upper right panels in Fig 7A).

We next investigated the phase behavior of $Ca^{2+}$-cAMP in these Turing pattern-based nanodomains. To do so, we used the Turing pattern in the lower middle panel in Fig 7A as the initial condition of AKAP/AC nanodomains (left panel in Fig 7B; also see Turing pattern-based nanodomain subsection in Methods for details). In this case, we have experimentally informed patterns but do not know the concentrations of the species. Since the simple ODE model indicates biochemical activity is important, we used the concentration of AC as free parameters and conducted a parameter variation. During this parameter variation, the largest concentration of AKAP on the AKAP/AC nanodomain was fixed to 9600 copies /$\mu m^2$ to

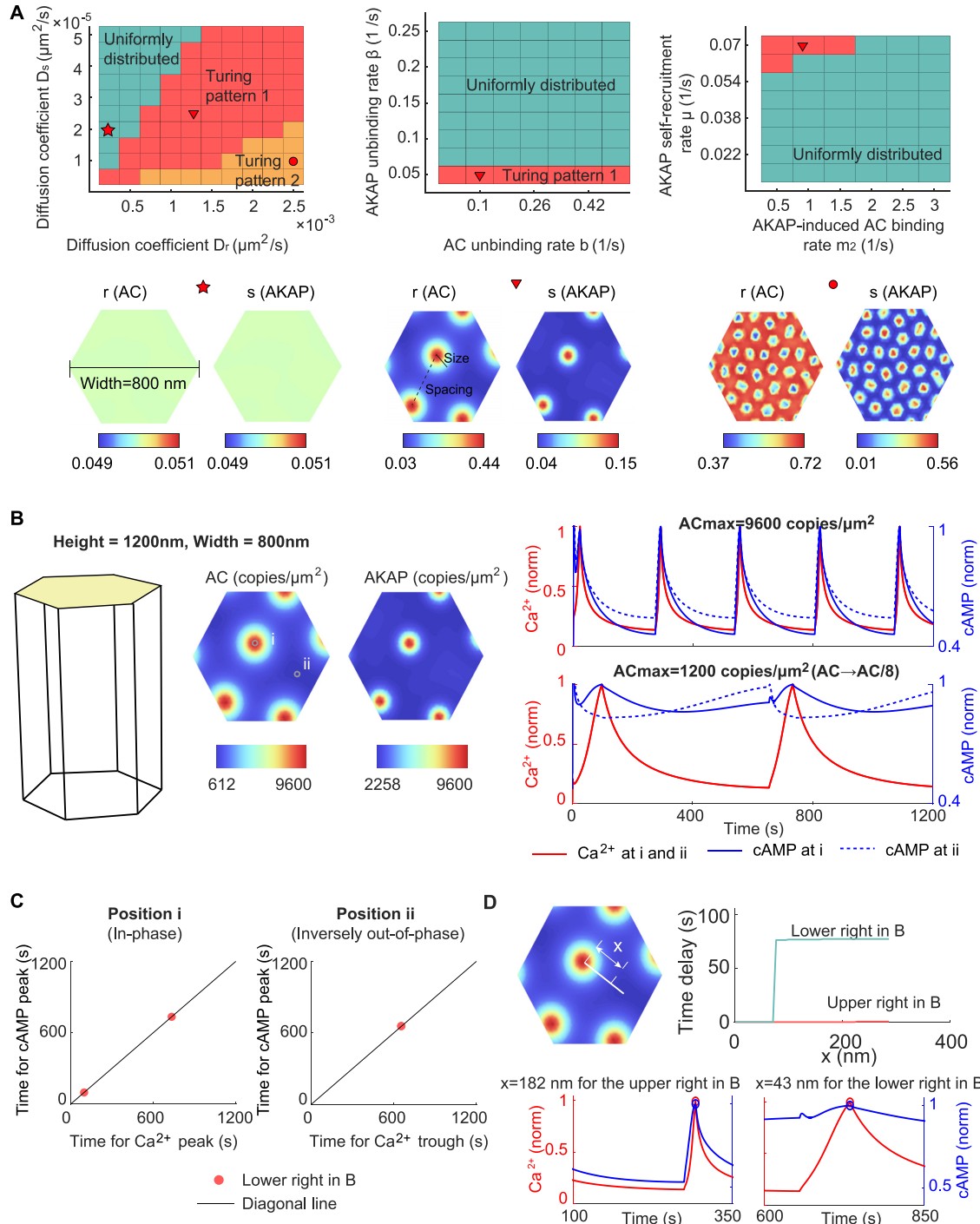

**Fig 7. AKAP/AC nanodomain formation can be explained by Turing patterns.** (A) The phase diagram of Turing pattern for parameters $v_r$ and $v_s$ (upper left panel), $\beta$ and $b$ (upper middle panel), and $m_2$ and $\mu$ (upper right panel). Three patterns occur in the parameter space: the first is that the AC (denoted by $r$) and AKAP (denoted by $s$) are uniformly distributed; the second is the Turing pattern where the AKAP and AC are co-localized, which is denoted as Turing pattern 1; the third is also a Turing pattern but the AC level is high outside the AKAP cluster, which is referred to as Turing pattern 2. Typical distributions of AC and AKAP for these three types of pattern are shown in lower panels. The marker over the plot indicates the value of $D_r$, $D_s$, $\beta$, $b$, $m_2$ and $\mu$. (B) The dynamics of $Ca^{2+}$ and cAMP on (i) or outside AKAP/AC nanodomains (ii) when initial conditions of AC and AKAP are re-scaled from the lower middle panel in (A). When the maximal concentration of AC is same as that in Fig 6, the cAMP oscillates in phase with $Ca^{2+}$ on and outside AKAP/AC nanodomains (upper right panel); when the AC level is one eighth of that in Fig 6, the cAMP oscillates in phase with $Ca^{2+}$ on AKAP/AC nanodomains but out of phase with $Ca^{2+}$ outside AKAP/AC nanodomains (lower right panel). (C) The in-

phase and inversely out-of-phase oscillations in (B). (D) The time delay in (B) as a function of the distance $x$ to the center of AKAP/AC nanodomains.

ensure the consistency with simulation results in Fig 6. The first rescaling set the largest concentration of AC on the AKAP/AC nanodomain to 9600 copies /$\mu m^2$, which was used in Fig 6. Given this re-scaling, the in-phase cAMP oscillation exists not only on AKAP/AC nanodomains but also outside AKAP/AC nanodomains (upper right panel in Fig 7B). This is consistent with our previous finding of in-phase oscillations on the AKAP/AC nanodomain but not consistent with the inversely out-of-phase oscillations outside the AKAP/AC nanodomain. Next, we decreased the concentration of AC to 12.5% of the previous value, and thus the largest AC concentration on the AKAP/AC nanodomain becomes 1200 copies /$\mu m^2$. We found the emergence of inversely out-of-phase cAMP oscillation outside AKAP/AC nanodomains (lower right panel in Fig 7B, 7C and 7D). Thus, our model predicted that Turing pattern 1 combined with the ratio of AC to AKAP is critical in regulating the phase behavior of $Ca^{2+}$-cAMP. This prediction demonstrates the importance of both pattern formation and biochemical activity in $Ca^{2+}$-cAMP oscillation.

## 2.7 Prediction of the time delay from a realistic distribution of AKAP/AC nanodomains

Finally, we extended our analysis to the realistic distribution of AKAP/AC nanodomains from AC8 STORM data in [32]. This analysis allows us to relax the previous assumptions of the nanodomains as either Gaussian distribution or Turing pattern and test our models in experimentally observed nanodomain distributions. To do this, we selected one square area with a length of 1000 nm from the STORM data (left and middle panels in Fig 8A); and then smoothed the STORM data in the square area by convoluting with a Gaussian function whose FWHM (full width at half maximum) is 60 nm (right panel in Fig 8A). After these two processes, we obtained a smooth distribution of AC8 on a square area with a length of 1000 nm. Next, the AC8 distribution in a hexagonal region denoted by the white dashed line (right panel in Fig 8A) is used as the initial distribution of AC in our simulations. Similar to the case with Turing pattern-based AKAP/AC nanodomain, the realistic concentration of AC is unknown. As suggested by the model results with Turing pattern-based AKAP/AC nanodomain, the AC activity plays an important role in the phase behavior of $Ca^{2+}$-cAMP, and thus we also tested two values of rescaled AC here: the first is that the largest value of AC concentration on the AKAP/AC nanodomain is rescaled to 9600 copies/$\mu m^2$ (upper right panel in Fig 8B), and the second is 4800 copies/$\mu m^2$ (lower right panel in Fig 8B). Additionally, the initial distribution of AKAP is always rescaled to ensure the highest concentration on AKAP/AC nanodomain of 9600 copies/$\mu m^2$.

The two different rescalings of the AC activity result in distinct phase behavior. The first rescaling shows in-phase $Ca^{2+}$-cAMP oscillation on and outside the AKAP/AC nanodomains (upper right panel in Fig 8B). For the second rescaling of 4800 copies/ m 2, the in-phase cAMP oscillation on AKAP/AC nanodomains and inversely out-of-phase cAMP oscillation outside the AKAP/AC nanodomains arise again (lower right panel in Fig 8B, 8C and 8D). These in-phase and inversely out-of-phase cAMP oscillations also appear when smoothing the AC STORM data with a Gaussian function whose FWHM is 30 nm (S5 Fig), because the AC activity with a 30 nm FWHM is less than or equal to that with 60 nm FWHM at any location. Therefore, our model predicts that the AC activity is essential for the phase difference between $Ca^{2+}$-cAMP even for a realistic distribution of AKAP/AC nanodomains.

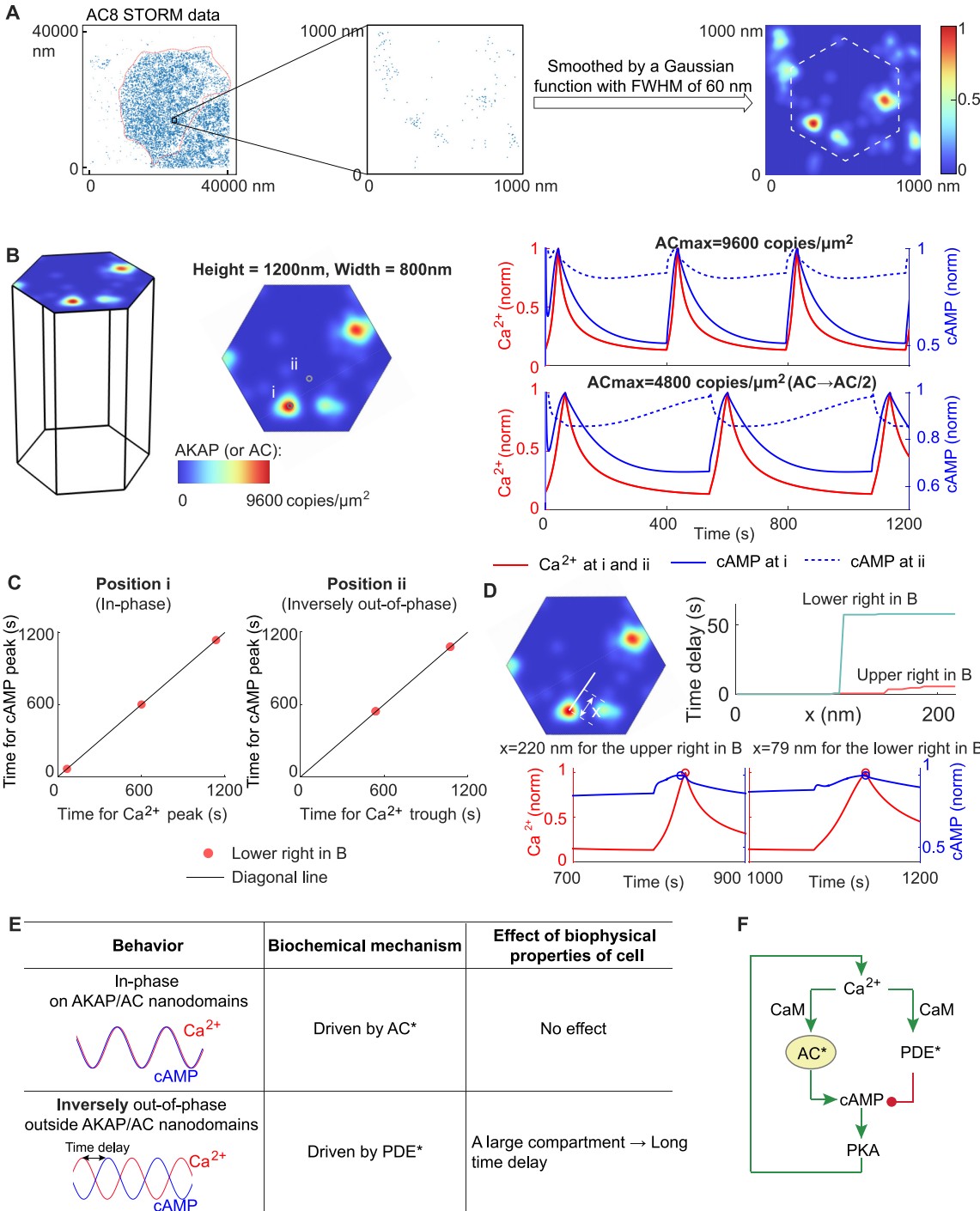

**Fig 8. Prediction of the time delay from a realistic distribution of AKAP/AC nanodomains.** (A) Extraction of AC8 data from STORM image. The AC8 STORM data (left panel) is from [32], where the red dashed line indicates the boundary of a single cell. We selected an area with 1000 nm length and 10000 nm width (middle panel), then smoothed the STORM data in this small area by a Gaussian function with FWHM (full width at half maximum) of 60 nm, and finally obtained a smooth distribution of AC8 on the small area (right panel). We chose a hexagon indicated by the white dashed line as the distribution of AC in the following simulations. (B) The dynamics of $Ca^{2+}$ and cAMP on the AKAP/AC nanodomain (i) and outside the AKAP/AC nanodomain (ii) when the initial distribution of AC is from (A). The initial distribution of AKAP is set to be the same as that of AC. Two cases are studied: the AC has the same maximal values as before (i.e., 9600 copies/$\mu m^2$) (upper right panel); the AC level is halved (lower right panel). (C) The in-phase and inversely out-of-phase oscillations in the lower right panel in (B). (D) The time delay as a function of the distance to the AKAP/AC nanodomain. (E) Summary of $Ca^{2+}$-cAMP oscillation, underlying biochemical mechanisms, and the effect of the

biophysical properties of the cell. (F) The core motif of the $Ca^{2+}$/cAMP circuit—incoherent feedforward loop from $Ca^{2+}$ to cAMP and the feedback from cAMP to $Ca^{2+}$.

## 3 Discussions

Signaling nanodomains and biomolecular condensates are ubiquitous within the cell. They regulate signaling transduction pathways by sequestering biochemical reactions. However, how these signaling aggregates precisely regulate intracellular pathways remains elusive, especially when taking the nonuniform size and distribution of these signaling aggregates in the cell into consideration. Here, we looked at the intersection of AKAP/AC nanodomain and the $Ca^{2+}$/cAMP pathway in detail and studied the underlying biochemical mechanisms and potential biophysical effects.

We developed a 3D reaction-diffusion model and different types of preexisting AKAP/AC nanodomains to study how AC and PDE activities, compartment size, and pattern of AKAP/AC nanodomains affect the phase shift between $Ca^{2+}$ and cAMP oscillation. Through simulating the 3D reaction-diffusion model with a Gaussian distribution-modeled AKAP/AC nanodomain, we revealed the paradoxical role of AC and PDE: active AC and active PDE drive in-phase oscillation on AKAP/AC nanodomains and out-of-phase oscillation outside AKAP/AC nanodomains, respectively (first and second columns in Fig 8E). It should be noted that the out-of-phase oscillation exhibits the same $Ca^{2+}$ trough time and cAMP peak time, which is validated by the close match of $Ca^{2+}$ trough time and cAMP peak time in the experiment. Such paradoxical role of AC and PDE in regulating the phase shift between $Ca^{2+}$ and cAMP can be explained by a simple incoherent feedforward loop and predicted by varying AC activity in the 3D reaction-diffusion model. Furthermore, the important role of AC in regulating phase delay between $Ca^{2+}$ and cAMP has been demonstrated by knocking down AC, overexpressing AC, or disturbing the interaction between AKAP and AC (Figure 2 (d-e) and Figure 5(b) in [32]). Moreover, these in-phase and inversely out-of-phase behaviors persist in most cases, but the time delay for the inversely out-of-phase behavior depends on the compartment size (third column in Fig 8E). Furthermore, the simulation results with Turing pattern-based and AC STORM data-based AKAP/AC nanodomains predicted the important role of nanodomain pattern and AC activity in coordinating the phase between $Ca^{2+}$ and cAMP.

Interestingly, we found that the phase shift between $Ca^{2+}$ and cAMP is controlled by an incoherent feedforward loop: the activating $Ca^{2+}$-AC-cAMP pathway and the inhibiting $Ca^{2+}$-PDE-cAMP pathway (Fig 8F). Although the cAMP can regulate the $Ca^{2+}$ level by a feedback loop through PKA (Fig 8F), it seems that the difference in oscillatory dynamics of cAMP at distinct cellular locations does not result in diverse $Ca^{2+}$ dynamics due to the fast diffusion of $Ca^{2+}$. Several well-known biological functions of the incoherent feedforward loop include achieving adaptation [59, 60, 71], regulating structural plasticity [61], processing oscillation [55], detecting fold-change [57], generating non-monotonic gene input functions [56], and facilitating adaptive tuning of gene expression [58], but the phase regulation of second messengers is first discovered in this work to our knowledge. Since the cell decodes information involved in second messengers to regulate gene transcription [14, 31, 72], the incoherent feedforward loop may play a broad role in gene expression. In addition, a finite phase difference between cAMP and $Ca^{2+}$ has been frequently observed in experiments [46, 53], suggesting the limited contribution of AKAP/AC nanodomain in governing the average cAMP and $Ca^{2+}$ dynamics within the cell.

The exploration of mechanisms of in-phase and inversely out-of-phase behavior in this work may help to understand the synchronization in oscillatory systems. The synchronization, which is widely observed in circadian clocks [73], long-term memory [74, 75], and collective behavior in cell populations [76, 77], is referred to as the locked phase for several interacting biological oscillators. Depending on the phase difference, the synchronization includes in-phase synchronization, anti-phase (or inversely out-of-phase) synchronization, and synchronization with an arbitrary phase shift. Therefore, if we regarded the $Ca^{2+}$ and cAMP as two biological oscillators, their in-phase and inversely out-of-phase behavior are two types of synchronization. Our numerical results suggest that improving the activation strength from one oscillator to the other ($v$ in Eq (2)) can only lead to a sudden change from inversely out-of-phase synchronization to in-phase synchronization rather than a continuous change. Furthermore, a continuous change in phase difference can be achieved by tuning the basal inhibition rate of the downstream oscillator ($k_2$ in the main text), and the large rate results in the in-phase or inversely out-of-phase synchronization.

Our work demonstrates the role of biophysical properties of the cell in $Ca^{2+}$-cAMP oscillation, which is another evidence that the biophysical properties of cells encode the signaling information. One important biophysical property of cells is cell shape. For example, cell aspect ratio affects cytosolic calcium levels in vascular smooth muscle cells [78]; the structure and the shape of the dendrite spine regulate AMPAR dynamics and energy landscape [79, 80]; the cell eccentricity encodes the information of growth factor receptor pathways [62]. While most studies focused on the change in the increase or decrease of intracellular signaling levels, our work observed the change in period and phase for oscillatory signaling molecules. Apart from the cell shape, the distribution of AKAP/AC nanodomains also plays a role in $Ca^{2+}$-cAMP oscillation. This result implies the importance of the signaling hubs, which has been revealed in many biological systems [81–84]. The major function of signaling hubs is to sequester the chemical reactions to ensure their occurrence in the right locations, and AKAP/AC nanodomains also exhibit similar behavior: the activation of AC can only happen on AKAP/AC nanodomains, and thus guarantees the high value of cAMP on AKAP/AC nanodomains.

The phase behavior of $Ca^{2+}$-cAMP in pancreatic β cell may be crucial to insulin secretion. The cAMP not only affects the $Ca^{2+}$ level as modeled in this work but also regulates PKA and guanine exchange factor Epac, two types of signaling molecules that contribute to secretory granule trafficking and exocytosis. On the one hand, both PKA and Epac improve the mobilization of secretory granules, resulting in fast replenishment of readily releasable granules [85–87]. On the other hand, several proteins involved in the exocytosis process have been identified as substrates of PKA or interacted with Epac, such as SNAP-25 and Snapin [88–91]. Moreover, AKAP nanodomains can anchor both PKA and Epac [92, 93], and thus may drive the oscillation of PKA and Epac during pulsatile insulin secretion. In addition to the potential effect in insulin secretion, the out-of-phase $Ca^{2+}$-cAMP behavior in pancreatic β cell encode the information in extracellular signals [53].

The AKAP/AC nanodomains mediate the phase difference between $Ca^{2+}$ and cAMP. Such nanodomains have been extensively found in the intracellular space as well as biomolecular condensates [94–97]. These signaling nanodomains and biomolecular condensates are able to directly regulate cellular processes, and dysregulations of these complexes are likely indicated as potential drivers of oncogenic activity. For example, the tumor suppressor SPOP (speckle-type POZ protein) granules are disrupted in the presence of SPOP mutations that promote prostate cancer progression [98, 99]; the stress granules, which are composed of mRNA transcripts, RNA-binding proteins, and translation machinery to prevent cell

apoptosis from stressful conditions, are markedly upregulated in cancer cells, resulting in more stress-resistant cancer cells [100, 101]; the promyelocytic leukemia (PML) nuclear bodies have been found to contribute to the telomere lengthening in cancer cells, thus improving the replicative ability of cancer cells [102, 103]; the flat clathrin lattice (FCL) is located on the cell membrane, providing sustained signaling hubs for epidermal growth factor receptor (EGFR), and the disruption of FCL leads to a decreased level of active EGFR on the cell membrane [81].

In addition to its application in receptor nanodomains, the Turing model is commonly employed to understand mechanisms underlying a wide range of biological patterns such as sea shell patterns, hair follicle spacing, hydra regeneration, lung branching and digits formation [104–107]. With a simple interaction network incorporating a short-range positive feedback and a long-range negative feedback (i.e., a slowly diffusing activator and a rapidly diffusing inhibitor), the Turing model is able to reproduce diverse spatial patterns. For example, a simple Turing reaction-diffusion model consisting of the phosphorylated and unphosphorylated species of Polo-like-kinase 4 (PLK4), is used to explain how mother centrioles break symmetry to generate a single daughter [108]. When taking the intrinsic noise into account, the stochastic version of Turing model can further relax parameter constraints such as the ratio of activator–inhibitor diffusion coefficients, which is validated in a synthetic bacterial population where the signaling molecules form a stochastic activator–inhibitor system [109].

Although we used the Turing model to explain the formation of AKAP/AC nanodomains, it is not the only one that can generate the pattern. For example, the particle-based model [67–70], lattice model [110, 111], and Cahn–Hilliard equation [112–114] can also produce patterns. The Cahn–Hilliard equation is well-known as the model for phase separation, but recently it has been used to model the formation of nanodomains on cell membranes. We also note that the Turing pattern in this work is a qualitative match but not a quantitative one, since the Turing pattern was not exactly fitted to experimental data: the spacing between two domains and size (black lines in the lower middle panel in Fig 7A) are 365 nm and 118 nm, respectively, while those in the experiment are 341 nm and 105 nm. This may be because the parameter settings of mean-shift algorithm when identifying the AC clusters from AC STORM data are different, or because the staining for the STORM data is not perfect, which causes significant artificial variations in domain spacing and size. Moreover, the spacing between two adjacent nanodomains in Turing pattern-based nanodomains is fixed in our study, and thus the effect of interactions between two adjacent nanodomains on $Ca^{2+}$/cAMP oscillation behavior is not considered. According to the almost same time delay when the distance of two AKAP/AC nanodomains varies (S6 Fig; also see Interaction between two nanodomains subsection in Methods), we anticipate that the oscillation behavior is not affected by the distance between nanodomains under certain circumstances. In addition to the limitation in choosing the patterning model, another limitation in our work is that we did not consider the existence of RIα (a subunit of PKA) condensates in the cytosol, which can accumulate cAMP locally [84]; future work is expected to incorporate RIα condensates to fill the knowledge gap. Another limitation is that we neglected the biological noise in the $Ca^{2+}$/cAMP circuit, because the number of molecules can be small within the cell. For example, there are 15 to 150 individual AC8 molecules in one AKAP/AC nanodomain [32]. We anticipate the emergence of novel behavior and mechanisms when modifying the deterministic model to a stochastic one [115–119]. Despite these limitations, our findings of $Ca^{2+}$-cAMP interacting through an incoherent feedforward loop to regulate their in and out-of-phase oscillations on AKAP/AC nanodomains sheds light on how spatial arrangement of molecules can regulate temporal dynamics of second messengers in different cellular locations.

## 4 Methods

### 4.1 Reaction-diffusion model

To investigate the underlying biochemical mechanism and the effect of biophysical properties of the cell as mentioned in Fig 1B, we used the 3D reaction-diffusion model in [32] to describe the dynamics of $Ca^{2+}$, cAMP, and other molecules that interact with $Ca^{2+}$ and cAMP. Next, we introduced this model and listed the equations below.

**4.1.1 Compartment.** As mentioned before, though calcium and cAMP are mostly distributed in the cytosol, they both can interact with molecules on the cell membrane, such as AKAP and AC. To take these chemical reactions on cell membrane into consideration, we chose a small compartment in one cell, that is, a hexagonal prism (Fig 1C), where the top surface (denoted by $\Gamma$) represents the cell membrane and the volume under the top surface (denoted by $\Omega$) the cytosol. Besides, the height and width of the compartment are set to be 600 nm and 400 nm respectively, which are consistent with those in [32]. However, in order to study the effect of the compartment size on the time delay between $Ca^{2+}$ and cAMP, the height and width may vary, which are labeled in figures and illustrated in the main text.

**4.1.2 Chemical reactions and governing equations.** For the $Ca^{2+}$-cAMP circuit, we considered 20 chemical species in total (Fig 1C). On the one hand, there are 13 chemical species in the cytosol (non-highlighted species in Fig 1C), including calcium, cAMP, CaM (calmodulin), PDE (Cyclic nucleotide phosphodiesterase), PKA (protein kinase A), and complexes of these molecules. On the other hand, there are 7 chemical species or quantities on the membrane (highlighted in yellow in Fig 1C), such as AC (Adenylyl cyclase), AKAP, and complexes of these molecules. These molecules in the cytosol and on the membrane have been proven to play crucial roles in regulating the $Ca^{2+}$-cAMP pathway [8, 120, 121]. For the full list of species, we refer the readers to S1 Table.

The dynamics of these 20 chemical species are governed by chemical reactions. The $Ca^{2+}$ in the cytosol can bind to CaM, leading to the activation of inactive PDE and inactive AC (v1–8). Since the inactive AC has been validated to be co-localized with AKAP by forming AKAP/AC nanodomains on the cell membrane [32], the activation of inactive AC caused by the $Ca^{2+}$/CaM complex is only distributed on the AKAP/AC nanodomain. Then, the active AC (denoted as $AC^*$) on AKAP/AC nanodomains can improve the synthesis of cAMP (v9), while the active PDE (denoted as $PDE^*$) degrades the cAMP in the cytosol (v10–13). Under the stimulus of cAMP, PKA in the cytosol is activated (v14–15). Another pathway to activate PKA is localized at the AKAP/AC nanodomain: AKAP binds to inactive PKA (v16–18) in the absence of cAMP, and these inactive PKA will also become active and dissociate with AKAP in the presence of cAMP (v19–20). Those active PKA can increase the synthesis of $Ca^{2+}$ (v23). $Ca^{2+}$ level is also increased via the influx across the plasma membrane (v22), and $Ca^{2+}$ regulates the $Ca^{2+}$ gated $K^+$ channel (v21). For the full list of chemical reactions, we refer the readers to S2 Table.

According to the type of reaction, these reactions can be categorized into two classes: the first type is the production, degradation, or binding event (solid arrows in Fig 1C), and the other is the regulatory reaction (dashed arrows in Fig 1C). For the first type, the species before and after the arrow are the reactant and product, respectively. For example, $AC + Ca_2CaM$ CaM AC means that the AC can bind to $Ca_2CaM$ and thus form the complex CaM AC. However, for the second type, the species before the arrow regulates the chemical reaction instead of working as the reactant; for instance, the reaction v12 means that the active PDE enhances the cAMP degradation. Due to the meaning of each type, one key difference between these two types is that the species before the arrow will be consumed for the first type of reaction but not for the second type of reaction.

To describe the dynamics of 20 chemical species in the compartment, the 3D diffusion-reaction model in [32] is adopted, which is shown as follows:

$$
\begin{cases}
\begin{cases}
[\text{Ca}^{2+}]_t &= -2j_1 - j_2 - j_3 - 2j_7 + j_{23} + D_{\text{Ca}^{2+}}\nabla^2[\text{Ca}^{2+}] \\
[\text{CaM}]_t &= -j_1 + D_{\text{CaM}}\nabla^2[\text{CaM}] \\
[\text{Ca}_2\text{CaM}]_t &= j_1 - j_2 - j_6 + D_{\text{Ca}_2\text{CaM}}\nabla^2[\text{Ca}^{2+}] \\
[\text{Ca}_3\text{CaM}]_t &= j_2 - j_3 + D_{\text{Ca}_3\text{CaM}}\nabla^2[\text{Ca}_3\text{CaM}] \\
[\text{Ca}_4\text{CaM}]_t &= j_3 - j_8 + D_{\text{Ca}_4\text{CaM}}\nabla^2[\text{Ca}_4\text{CaM}] \\
[\text{PDE}]_t &= -j_6 - j_8 + D_{\text{PDE}}\nabla^2[\text{PDE}] \\
[\text{CaM}\cdot\text{PDE}]_t &= j_6 - j_7 + D_{\text{CaM}\cdot\text{PDE}}\nabla^2[\text{CaM}\cdot\text{PDE}] \\
[\text{PDE}^*]_t &= j_7 + j_8 + D_{\text{PDE}^*}\nabla^2[\text{PDE}^*] \\
[\text{cAMP}]_t &= -j_{10} - j_{11} - j_{12} - j_{13} - 2j_{14} - 2j_{15} \\
& \quad + D_{\text{cAMP}}\nabla^2[\text{cAMP}] \\
[\text{R}_2]_t &= j_{15} + D_{\text{R}_2}\nabla^2[\text{R}_2] \\
[\text{R}_2\text{C}]_t &= j_{14} - j_{15} + D_{\text{R}_2\text{C}}\nabla^2[\text{R}_2\text{C}] \\
[\text{R}_2\text{C}_2]_t &= -j_{14} + D_{\text{R}_2\text{C}_2}\nabla^2[\text{R}_2\text{C}_2] \\
[\text{PKA}]_t &= j_{14} + j_{15} + D_{\text{PKA}}\nabla^2[\text{PKA}]
\end{cases} & \text{in } \Omega \\[2pt]
\begin{cases}
[\text{AC}]_t &= -j_4 \\
[\text{CaM}\cdot\text{AC}]_t &= j_4 - j_5 \\
[\text{AC}^*]_t &= j_5 \\
[\text{AKAP}]_t &= -j_{16} - j_{17} - j_{18} \\
[\text{AKAP}-\text{R}_2]_t &= j_{18} + j_{20} \\
[\text{AKAP}-\text{R}_2\text{C}]_t &= j_{19} - j_{20} + j_{17} \\
[\text{AKAP}-\text{R}_2\text{C}_2]_t &= j_{16} - j_{19}
\end{cases} & \text{on } \Gamma
\end{cases}
\tag{5}
$$

where $[\cdots]$ denotes the concentration of species. The $\Gamma$ is the top surface of the compartment, and $\Omega$ is the volume under $\Gamma$ (see Fig 1C). The term on the right-hand side is the time derivative, and that on the left-hand side is composed of reaction parts and diffusion parts ($D_X \nabla^2[X]$, $X = \text{Ca}^{2+}, \text{CaM}, \cdots, \text{PKA}$). For reaction parts, the $j_i$ ($i = 1, 2, \ldots, 20, 22, 23$) represent the flux of reactions vi; (see S2–S4 Tables). Note that the flux of the production, degradation or binding event (solid arrows in Fig 1C) will lead to not only the consumption of reactants but also the production of products. However, the reaction that regulates other reactions (dashed arrows in Fig 1C) only affects the reaction rate while maintaining the concentration of regulator species. As for diffusion parts, the $D_X$ ($X = \text{Ca}^{2+}, \text{CaM}, \cdots, \text{PKA}$) denotes the diffusion coefficient of species $X$ (see S5 Table for exact values); the $\nabla^2$ is the Laplace operator representing $\frac{\partial^2}{\partial x^2} + \frac{\partial^2}{\partial y^2} + \frac{\partial^2}{\partial z^2}$; the $\nabla^2_\Gamma$ denotes the diffusion on the cell membrane $\Gamma$ and is defined as $\frac{\partial^2}{\partial x^2} + \frac{\partial^2}{\partial y^2}$.

As for the boundary condition, most of species are assumed to have the zero flux on the boundary $\Gamma$, and those with non-zero fluxes on $\Gamma$ are listed as follows:

$$
\begin{cases}
-D_{\text{Ca}^{2+}}\vec{n}\cdot\nabla[\text{Ca}^{2+}] & = & j_{22} - 2j_5, \text{ on } \Gamma \\
-D_{\text{Ca}_2\text{CaM}}\vec{n}\cdot\nabla[\text{Ca}_2\text{CaM}] & = & -j_4, \text{ on } \Gamma \\
-D_{\text{cAMP}}\vec{n}\cdot\nabla[\text{cAMP}] & = & j_9 - 2j_{19} - 2j_{20}, \text{ on } \Gamma \\
-D_{\text{R}_2}\vec{n}\cdot\nabla[\text{R}_2] & = & -j_{18}, \text{ on } \Gamma \\
-D_{\text{R}_2\text{C}}\vec{n}\cdot\nabla[\text{R}_2\text{C}] & = & -j_{17}, \text{ on } \Gamma \\
-D_{\text{R}_2\text{C}_2}\vec{n}\cdot\nabla[\text{R}_2\text{C}_2] & = & -j_{16}, \text{ on } \Gamma \\
-D_{\text{PKA}}\vec{n}\cdot\nabla[\text{PKA}] & = & j_{19} + j_{20}, \text{ on } \Gamma
\end{cases}
\tag{6}
$$

where $\vec{n}$ denotes the exterior normal vector to the membrane $\Gamma$. These non-zero fluxes result from the reactions whose reactant or product is located on the cell membrane $\Gamma$. For the six surfaces surrounding the compartment, periodic boundary conditions are applied. The bottom surface is assumed to have zero fluxes.

The initial condition for species in the cytosol is set to be uniformly distributed (see S6 Table for exact values). For those species on the membrane, their initial conditions are set to be 0 except for AKAP and AC (S6 Table). Since AKAP can form nanodomains [32], we used the following Gaussian distribution to model one AKAP nanodomain:

$$
[\text{AKAP}]|_{t=0} = A\frac{1}{\sqrt{2\pi}\sigma}exp\left(-\frac{1}{2}\frac{(x-x_0)^2 + (y-y_0)^2}{\sigma^2}\right), \quad (x,y) \in \Gamma
\tag{7}
$$

where $\sigma$ and $A$ are set to be 25 nm and 6.1446E11 copies/nm respectively. The $x_0$ and $y_0$ denote the x and y coordinates of the nanodomain center. For example, in Fig 1D where the AKAP/AC nanodomain is at the center of the cell membrane $\Gamma$, the $x_0$ and $y_0$ are both zero. Moreover, the initial concentration of AC on the cell membrane is set to be the same as that of AKAP, unless otherwise specified. Besides, the derivative of the concentration with respect to time for all species is set to be 0 at time 0 (S6 Table).

**4.1.3 Voltage module and governing equations.** As we mentioned above, the $\text{Ca}^{2+}$ level can be increased by the influx via $\text{Ca}^{2+}$ channel (v22). Therefore, we used a voltage module where the $\text{Ca}^{2+}$ channel is modeled by an electrical conductance. In addition to $\text{Ca}^{2+}$ channel, $\text{Ca}^{2+}$ gated $\text{K}^+$ channel, $\text{K}^+$ channel, and the leak channel are also included. Currents for ion channels are represented by $I_i$, $i = Ca, KCa, K$, and $L$. The cell membrane is modeled by a capacitor with capacitance $C_m$. In the Hodgkin-Huxley framework, the membrane current is the sum of all the contributions from ion channels, i.e., $C_m\frac{dV}{dt} = -I_{KCa} - I_{Ca} - I_L - I_K$, where $V$ is membrane voltage. Besides, the membrane voltage $V$ is assumed to diffuse on the cell membrane with the diffusion coefficient $D_V$. Thus, the dynamics of membrane voltage $V$ on $\Gamma$ is governed by

$$
\begin{aligned}
V_t & = \frac{1}{C_m}(-I_{KCa} - I_{Ca} - I_K - I_L) + D_V\nabla^2_\Gamma V \\
\\
& = \frac{1}{C_m}\left(-g_{KCa}\frac{[Ca^{2+}]}{[Ca^{2+}] + K_{KCa}}(V - E_{KCa}) - g_{Ca}\frac{1 + tanh\left(\frac{V - v_1}{v_2}\right)}{2}(V - E_{Ca}) - g_K w(V - E_K) - g_L(V - E_L)\right) + D_V\nabla^2_\Gamma V,
\end{aligned}
\tag{8}
$$

where $\frac{[Ca^{2+}]}{[Ca^{2+}]+K_{KCa}}$, $\frac{1+tanh\left(\frac{V-v_1}{v_2}\right)}{2}$, and $w$ are the steady-state fraction of open $Ca^{2+}$ gated $K^+$, $Ca^{2+}$, and $K^+$ channels, respectively. For the specific $i$ ($i$ = KCa, Ca, K, L) channel, $g_i$ and $E_i$ denotes the conductance and reversal potential, respectively. The governing equation for the $K^+$ channel open probability $w$ on $\Gamma$ is as follows:

$$w_t = \phi \frac{w_\infty - w}{\tau} \tag{9}$$

where $\tau = \frac{1}{cosh\left(\frac{V-v_3}{2v_4}\right)}$ is the time constant, $w_\infty = \frac{1}{2} + \frac{1}{2}tanh\left(\frac{V-v_3}{v_4}\right)$ is the steady-state fraction of

open $K^+$ channel. $\phi$ is the factor that controls the relative time scales of $V$ and $w$. See S4 Table for the value of constants $K_{KCa}$, $v_1$, $v_2$, $v_3$, $v_4$, $\phi$, $g_i$, and $E_i$ ($i$ = KCa, Ca, K, L).

**4.1.4 Coupling between chemical reactions and voltage module.** The coupling between chemical reactions and voltage module is achieved by two reactions: the regulation from $Ca^{2+}$ concentration to the $Ca^{2+}$ gated $K^+$ channel (v21), and the $Ca^{2+}$ flux through $Ca^{2+}$ channel (v22). For the reaction v21, $Ca^{2+}$ concentration determines the steady-state fraction of open $Ca^{2+}$ gated $K^+$ channel in a Hill function form (See S4 Table for parameter values), i.e., $\frac{[Ca^{2+}]}{[Ca^{2+}]+K_{KCa}}$ in the Eq (8). For the reaction v22, the $Ca^{2+}$ flux through $Ca^{2+}$ channel is modeled by $j_{22} = (-\alpha I_{Ca} - v_{LPM}[Ca^{2+}])(1 + k_{PKAV}[PKA])$ (See S4 Table for parameter values), which affects the boundary condition of $Ca^{2+}$ in the Eq (6).

**4.1.5 Numerical simulations.** To simulate the dynamics of all molecules in the above mathematical model, we used COMSOL Multiphysics7.1 to numerically solve Eqs (5), (6), (8) and (9). The COMSOL files are available at https://github.com/RangamaniLabUCSD/Qiao_et_al_cAMP-Calcium-on-AKAP-AC-nanodomains.

## 4.2 Simple ODE model

To simplify the Eq (1), the time $t$ is multiplied by $m$, and $v$ and $k_1$ are normalized by $m$. For the simplicity of notation, we kept the notation of parameters and obtained the following equation:

$$\frac{dx}{dt} = vS(t/m) + k_1 - [S(t/m) + k_2]x.$$

It can be seen that the parameter $m$ only influences the period of the signal. Since the period of $Ca^{2+}$ is the same in the entire compartment, we can fix the value of $m$ to 1 and get the Eq (2).

Next, we will derive the solution of the Eq (2). Let $\tilde{x} = x - \frac{k_1}{k_2}$, then the equation for $\tilde{x}$ is shown as follows:

$$\frac{d\tilde{x}}{dt} = \left(v - \frac{k_1}{k_2}\right)S(t) - \tilde{x}[S(t) + k_2].$$

By using the variation of parameters method, we can obtain the expression for $\tilde{x}$:

$$\tilde{x}(t) = \left(v - \frac{k_1}{k_2}\right)e^{-I(t)-k_2t}\int_0^t S(s)e^{I(s)+k_2s}ds + \tilde{x}(0)e^{-I(t)-k_2t},$$

where $I(t)$ is $\int_0^t S(s)ds = 1 - cos(t) + 1.1t$. Note that $\tilde{x} = x - \frac{k_1}{k_2}$, and thus the $x(t)$ can be

written as follows:

$$x(t) = \left(v - \frac{k_1}{k_2}\right) e^{-I(s)-k_2 t} \int_0^t S(s) e^{I(s)+k_2 s} ds + \left(x(0) - \frac{k_1}{k_2}\right) e^{-I(t)-k_2 t} + \frac{k_1}{k_2}.$$

When $t$ goes to infinity, the term $\left(x(0) - \frac{k_1}{k_2}\right) e^{-I(t)-k_2 t}$ will disappear because $e^{-I(t)-k_2 t}$ goes to zero. Therefore, the expression of $x(t)$ when $t$ goes to infinity is written as follows:

$$x(t) = \left(v - \frac{k_1}{k_2}\right) e^{-I(s)-k_2 t} \int_0^t S(s) e^{I(s)+k_2 s} ds + \frac{k_1}{k_2}.$$

which is exactly the Eq (3).

Then, we analyzed how the peak and trough time of $x(t)$ depends on kinetic parameters $v$, $k_1$, $k_2$ and the signal $S(t)$. For simplicity, we use $F(t)$ to denote $e^{-I(t)-k_2 t} \int_0^t S(s) e^{I(s)+k_2 s} ds$. Then, we used $t_{max,x}$ to denote the peak time of $x(t)$, and the following equations for $t_{max,x}$ hold:

$$x'(t_{max,x}) = 0, \quad x''(t_{max,x}) < 0.$$

By replacing $x(t)$ with the Eq (3), we obtain:

$$F'(t_{max,x}) = 0, \quad \left(v - \frac{k_1}{k_2}\right) F''(t_{max,x}) < 0.$$

Therefore, if $v < \frac{k_1}{k_2}$, the peak time of $x(t)$ is the trough time of $F(t)$; if $v > \frac{k_1}{k_2}$, the peak time of $x(t)$ is the peak time of $F(t)$. This explains the reason for only two values of time delay when increasing the activation strength $v$ and the transition point $v = \frac{k_1}{k_2}$ in Fig 4D. Due to the close relation between $x(t)$ and $F(t)$, we then analyzed the peak and trough time of $F(t)$. Recall that $I(t) = \int_0^t S(s) ds = 1 - cos(t) + 1.1t$, and thus the $F(t) \approx e^{-k_2 t} \int_0^t S(s) e^{k_2 s} ds$ due to $I(t) \ll k_2 t$ when $k_2 \rightarrow +\infty$. We integrated the $e^{-k_2 t} \int_0^t S(s) e^{k_2 s} ds$ by parts, and then let $t$ goes to infinity, leading to the following equation for $F(t)$:

$$F(t) \rightarrow \frac{S(t)}{k_2}, \text{when } t \rightarrow +\infty$$

Thus, the peak and trough time of $F(t)$ are the same as the signal $S(t)$. Combined with the relation between $x(t)$ and $F(t)$, we obtained that if $k_2$ is large enough, the peak time of $x(t)$ is the tough time of $S(t)$ for $v < \frac{k_1}{k_2}$ and is the peak time of $S(t)$ for $v > \frac{k_1}{k_2}$, which is consistent with results in Fig 4D(iii).

## 4.3 Turing pattern-based nanodomain

In order to use the Turing pattern as the initial condition of AKAP/AC nanodomain, we assumed that the formation of AKAP/AC nanodomains is much faster than reactions in the Ca²⁺-cAMP circuit, and thus we directly used the Turing pattern in the lower middle panel in Fig 7A. Note that we did not couple Eqs (4) into (5). We also assumed that the diffusion coefficients of AKAP and AC are zero once the Turing pattern is formed, corresponding to the zero diffusion terms in the equations for AKAP and AC in the 3D reaction-diffusion model.

## 4.4 Interaction between two nanodomains

We investigated how the distribution of AKAP/AC nanodomains affects the phase difference between Ca²⁺-cAMP. We extended our model to include two AKAP/AC nanodomains on the

cell membrane but with three distinct distances 240 nm (S6 Fig), 320 nm, and 400 nm. We found that the in-phase cAMP oscillation on the AKAP/AC nanodomains and inversely out-of-phase cAMP oscillation outside the AKAP/AC nanodomains still hold (S6 Fig). Furthermore, the time delay for the inversely out-of-phase case is always near 75 seconds although the distance between two AKAP/AC nanodomains is distinct (S6 Fig). This result suggests that the distance between two AKAP/AC nanodomains has no effect on the time delay.

## Supporting information

**S1 Table. Chemical species or quantities.**
(PDF)

**S2 Table. Chemical reactions.**
(PDF)

**S3 Table. Voltage module and related reactions.**
(PDF)

**S4 Table. Kinetic parameters (from [32];* means that the parameters are modified according to surface/volume relationships).**
(PDF)

**S5 Table. Diffusion coefficients (from [32]; values are estimated from the biologically plausible range.).**
(PDF)

**S6 Table. Parameters related to initial conditions (from [32]).**
(PDF)

**S7 Table. Kinetic parameters in the Turing model (from [48]).**
(PDF)

**S1 Fig. The non-normalized dynamics of species in the presence of one AKAP/AC nanodomain and in the absence of AKAP/AC nanodomain.** (A) The non-normalized dynamics of species for the simulation in Fig 2A. (i) The simulation domain and the initial condition of AC and AKAP. This plot is the same as that in Fig 2A. (ii) The non-normalized dynamics of $Ca^{2+}$ and cAMP at the center (solid line; location i) or edge (dashed line; location ii) of the cell membrane. The dynamics of $Ca^{2+}$ and cAMP are shown in red and blue, respectively. (iii-vi) The non-normalized dynamics of membrane voltage (iii), PDE (iv), active AC (v), and PKA (vi) (solid line; location i) or edge (dashed line; location ii) of the cell membrane. (B) Same plots as (A) except that the AKAP/AC nanodomain does not exist.
(EPS)

**S2 Fig. The non-normalized dynamics of species in the absence of AKAP/AC nanodomain and PDE.** (A) The simulation domain where the AKAP/AC nanodomain and PDE do not exist. The absence of PDE is achived by setting the time derivative of inactive PDE, the complex of CaM and PDE, active PDE to 0. (B-G) The non-normalized dynamics of $Ca^{2+}$ (B), cAMP (C), membrane voltage (D), PDE (E), active AC (F), and PKA (G) for the simulation domain in A.
(EPS)

**S3 Fig. The effect of membrane curvature on the time delay.** (A-B) The dynamics of $Ca^{2+}$ and cAMP for inward membrane (A) and outward membrane (B). Only two locations are considered: on the center of AKAP/AC nanodomains (i), and at the edge of the membrane (ii).

(C) The in-phase and inversely out-of-phase cAMP behavior in (A-B). (D) The time delay of peaks between $Ca^{2+}$ and cAMP as a function of the distance $x$ in (A-B). The $x$ denotes the distance from the center to the edge of the cell membrane.
(EPS)

**S4 Fig. Cluster distance and size for AC STORM data.** (A) one sample of AC STORM data and the distribution of the cluster distance and size for a $10\mu m \times 10\mu m$ region. (i) One sample of AC STORM data. (ii) The magnified plot of the region in the red box in (i). Each red circle labels one cluster: the center of the circle is calculated from the mean-shift algorithm [122, 123], and the radius of the circle is defined as the value such that the circle can include 95% points of a specific cluster. (iii) The histogram of nearest-neighbor distances in (ii). The mean of the nearest-neighbor distances is indicated in the upper right corner. (iv) The histogram of cluster sizes. The cluster size is defined as the radius of red circles in (ii). The mean cluster size is indicated in the upper right corner. (B-C) Same plot as (A) but for another AC STORM data sample.
(EPS)

**S5 Fig. Prediction of the time delay from a realistic distribution of AKAP/AC nanodomains but with FWHM (full width at half maximum) of 30 nm.** (A) The dynamics of $Ca^{2+}$ and cAMP on the AKAP/AC nanodomain (i) and outside the AKAP/AC nanodomain (ii) when the initial distribution of AC is obtained in a similar way as Fig 8A except the FWHM of 30 nm. (B) The in-phase and inversely out-of-phase oscillations. (C) The time delay as a function of the distance to the AKAP/AC nanodomain.
(EPS)

**S6 Fig. Time delay for the inversely out-of-phase $Ca^{2+}$-cAMP oscillation is not affected by the distance between two AKAP/AC nanodomains.** (A) The $Ca^{2+}$ and cAMP dynamics where two AKAP/AC nanodomains exist on the cell membrane with a distance of 240 nm. Dynamics on one AKAP/AC nanodomain (i) and at the edge of the compartment (ii) are shown. (B-C) Same plots as (A) except for the distance between two AKAP/AC nanodomains (320 nm in B and 400 nm in C). (D) The in-phase and inversely out-of-phase oscillations in (A-C). (E) The time delay in (A-C) as a function of the distance $x$ to the center of the membrane compartment. The lower panel shows the magnified view of the $Ca^{2+}$ and cAMP dynamics when $x$ is close to the critical value, whose definition is mentioned in the caption of Fig 6E.
(EPS)

## Author Contributions

**Conceptualization:** Jin Zhang, Padmini Rangamani.

**Data curation:** Lingxia Qiao, Michael Getz, Ben Gross, Brian Tenner.

**Formal analysis:** Lingxia Qiao.

**Methodology:** Michael Getz, Ben Gross, Brian Tenner.

**Supervision:** Jin Zhang, Padmini Rangamani.

**Writing – original draft:** Lingxia Qiao, Michael Getz, Ben Gross, Brian Tenner, Jin Zhang, Padmini Rangamani.

**Writing – review & editing:** Lingxia Qiao, Michael Getz, Ben Gross, Brian Tenner, Jin Zhang, Padmini Rangamani.

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
