## [Decision Letter · Decision Letter 0]

7 May 2024

Dear Dr. Rangamani,

Thank you very much for submitting your manuscript "Spatiotemporal orchestration of Ca2+-cAMP oscillations  on AKAP/AC nanodomains is governed by an incoherent feedforward loop" for consideration at PLOS Computational Biology.

As with all papers reviewed by the journal, your manuscript was reviewed by members of the editorial board and by several independent reviewers. In light of the reviews (below this email), we would like to invite the resubmission of a significantly-revised version that takes into account the reviewers' comments.

We cannot make any decision about publication until we have seen the revised manuscript and your response to the reviewers' comments. Your revised manuscript is also likely to be sent to reviewers for further evaluation.

Sincerely,

Marc R Birtwistle, PhD

Section Editor

PLOS Computational Biology

Stacey Finley

Section Editor

PLOS Computational Biology

Reviewer's Responses to Questions

**Comments to the Authors:**

Reviewer #1: Qiao et al study spatial oscillations of Ca2+ and cAMP around AKAP/AC

nanodomains. They develop a model for these oscillations, point out that

it includes an incoherent feedforward component, and explore what happens

at different positions from the nanodomain. They also suggest that Turing

dynamics may underlie nanodomain formation, and perform simulations on

experimentally motivated domain distributions.

General comments.

1. Time delay between CA2+ and cAMP peak.

The explanation of why there is a sharp transition in time delay with

distance is weak. From Fig 2C, it appears that it

is a subtle matter of swapping between two very similar peaks.

The normalization makes it unclear how large is the absolute amplitude.

In the abstracted model with a sinusoidal approximation, the amplitude

of the oscillation drops close to zero at the transition point.

2. The authors frequently use the phrase "perfectly out-of-phase Ca2+/cAMP

oscillation". See Figure 2. This is misleading. An out-of-phase signal is

generally interpreted to be pi radians away from the in-phase signal.

In this case the angular phase delay, and also the time delay are quite small.

The time delay is ~20s compared to the oscillation period of ~300 s. This

is a phase difference of only about 0.4 radians. The reason is that the

Ca rises rapidly from valley to peak, and the authors appear to consider

the valley to peak phase as perfectly opposite.

3. The authors also frequently refer to incoherent feedback, which arises

from the opposite effects of AC and of PDE on cAMP (formation and removal,

respectively). My reading of the provided data is that this is not a good

characterization of their feedback system. The reason is that AC is present

only in the nanodomain, and from what I can see (esp Fig 2), all the

relevant dynamics of the oscillatory system reside in the nanodomain.

With this interpretation, we have an oscillatory Ca and cAMP signal generated

at the nanodomain, with gradually declining levels of cAMP as one goes further

away. Ca remains relatively high because of its fast diffusion.

cAMP is then in phase with Ca near the nanodomain, but further away the

effect of PDE dominates, hence it is lower at the time of the Ca peak.

I note that the authors have plotted panel 2C and 2F as normalized cAMP,

so one does not get to see that actually cAMP is quite small further away

where PDE dominates. As mentioned above, the absolute values should also

be reported.

My interpretation is easily tested in a non-spatial model with the same

reaction scheme. I expect that the behavior will be just like the nanodomain

region of the current model.

Hence, to summarize the critiques here

- The feedback is not incoherent, it is entirely driven by the excitatory

arm of the pathway.

- The oscillatory dynamics are driven by the excitatory arm in the nanodomain

and this can be tested using a single-compartment model to replicate

the nanodomain properties.

- Any substantial inhibitory effect of PDE on cAMP occurs far from the

nanodomain, hence the PDE negative feedback effect on cAMP has little

effect on the total PKA activity.

- The cAMP levels in the region where PDE dominates are very small and unlikely

to modulate PKA significantly.

4. The authors then fold in a somewhat orthogonal question about formation of

nanodomains. They refer to STORM data, smooth it, and obtain small peaks which

they interpret as naondomains. Unfortunately they have used one sample,

which is not a reliable basis for interpreting nanodomain presence. Further,

they choose a Gaussian smoothing function of 60 nm without a clear rationale

for this length scale.

- Can the authors do the analysis in a statistically thorough manner, with

multiple samples and multiple biological replicates? This would be

the starting point for any interpretation on periodicity of the

nanodomains, which is a requisite to interpret them as Turing patterns.

- On the basis of this, can they obtain the wavelength of the proposed

Turing pattern?

- Further, can they use a larger sample set to obtain a population estimate

of proposed nanodomain size?

- I was looking to see an analysis that justified the strong assumption that

nanodomains were indeed formed through a Turing mechanism. My reading

is that the authors have come up with a potential Turing-like

mechanism for nanodomain formations, but I do not see experimental

support for either the existence of this mechanism nor the existence

of Turing like regular patterns. Hence at this stage this appears

like a hypothesis, which could be subject to future testing.

- Can the authors provide evidence that the AC8 distributions reported are

stable, that is, that these are long-lived nanodomains rather than

transients or experimental noise?

- There are potentially interesting interactions here between adjacent

nanodomains (if they are verified), and dependencies on their spacing.

These are explored briefly in Figure 7.

However the current single-sample analysis does not support such

generalization.

5. The authors state

"These reactions include negative feedback, incoherent feedback loop, and Hodgkin-Huxley model, and thus are able to generate oscillation."

The equation form for K channels is not in the HH form.

Also I do not see an equation for computing V, nor do I see plots for V

in the figures.

Minor points

Pg 52-53, table S3: "Voltage of K+ channel" is incorrect. I think the authors

mean to say reversal potential of K+ channel. Similarly for Ca2+ channel.

Reviewer #2: Spatiotemporal orchestration of Ca2+-cAMP oscillations on AKAP/AC nanodomains is governed by an incoherent feedforward loop" (PCOMPBIOL-D-24-00398)

Overall, the manuscript submitted by Dr. Padmini Rangamani and colleagues is well-written and of general interest to the calcium signaling/cAMP signaling community. The key incoherent circuit that can explain the changing of phase between Ca2+ and cAMP oscillations is a notable finding from the computational work. The key questions posed by the study are answered computationally.

Ref. 32 appears to have important data for the present study, but is only published as a presentation abstract. It seems that the citation for [32] is incorrect and should refer to https://elifesciences.org/articles/55013? Additionally, the novelty of the present computational work should be more expansively delineated from the eLife paper. It was not clear what is meant by “perfect property” of the out-of-phase behavior, as this out-of-phase activity seems to be discussed in the 2020 paper. The two studies should be explicitly compared and differentiated.

Example: “have been experimentally well studied in [32],”

Can specific downstream biological functions be assigned to the in-phase and out-of-phase behavior inside and outside the domains?

It would be helpful to provide a more specific rationale within the text for model parameters and calibration to save the reader time in referring to the previous studies.

Minor issues or unclear points:

Mention the specific factors in this summary statement: “but some of these factors can affect the time delay for the perfectly out-of-phase Ca2+/cAMP oscillation.”

The role of compartment height needs to be clearly explained in the text.

The term compartment size was somewhat confusing in relation to the biological structure and organization of the cell. Clarity in terminology would be helpful.

Reviewer #3: Please see attached comments.

**Have the authors made all data and (if applicable) computational code underlying the findings in their manuscript fully available?**

Reviewer #1: Yes

Reviewer #2: Yes

Reviewer #3: Yes

PLOS authors have the option to publish the peer review history of their article (what does this mean?). If published, this will include your full peer review and any attached files.

Reviewer #1: No

Reviewer #2: No

Reviewer #3: **Yes: **Andrew L. Krause
---

## [Decision Letter · Decision Letter 1]

16 Oct 2024

Dear Dr. Rangamani,

We are pleased to inform you that your manuscript 'Spatiotemporal orchestration of Ca2+-cAMP oscillations  on AKAP/AC nanodomains is governed by an incoherent feedforward loop' has been provisionally accepted for publication in PLOS Computational Biology.

Best regards,

Marc R Birtwistle, PhD

Section Editor

PLOS Computational Biology

Marc Birtwistle

Section Editor

PLOS Computational Biology

Reviewer's Responses to Questions

**Comments to the Authors:**

Reviewer #2: The Authors largely satisfied the suggestions and critiques raised in my first review. Unless I am missing something, "inversely" and "out-of-phase" is redundant, and could be replaced with "out-of-phase".

Reviewer #3: The authors have addressed all of my concerns, and I'm very happy to recommend this excellent paper for publication!

Reviewer #4: Qiao et al present a computational study investigating oscillations of Ca2+ and cAMP spatially around AKAP/AC nanodomains. They present a model based on an incoherent feedforward system that drives the oscillatory behavior of the secondary messengers, changing the phase behavior spatially. The model is well-characterized and generally validated with previous experimental data. This manuscript is well-written and is of general interest to the cAMP signaling research community. Further, the study is well motivated, as a better understanding of the oscillatory behavior of Ca2+ and cAMP in pancreatic cells could lead to a better understanding of diabetes disease dynamics. Finally, I find that the authors have adequately addressed the previous reviewers’ comments in the revised submission. Below are two minor issues that could add to the clarity of the paper.

1. In the first paragraph of results section 2.1, there appears to be a typo in referencing the reaction numbers presented in Table S2 and Figure 1C. In line 131, the activation of PDE is referenced in the text as reactions “(v4-5)”, but based on Table S2 & Figure 1C, I believe they should be “(v6-8)”. Similarly, in line 132, the activation of AC is referenced as “(v1-3 and v6-8)” but should be “(v1-5)”.

2. I found the presentation of Figure 2D to be a little confusing. It would be more clear if each x distance was plotted as an individual panel, rather than grouping x=0nm and x=200nm on a panel together and splitting x=49nm into a panel alone.

**Have the authors made all data and (if applicable) computational code underlying the findings in their manuscript fully available?**

Reviewer #2: Yes

Reviewer #3: Yes

Reviewer #4: Yes

PLOS authors have the option to publish the peer review history of their article (what does this mean?). If published, this will include your full peer review and any attached files.

Reviewer #2: No

Reviewer #3: **Yes: **Andrew L. Krause

Reviewer #4: No

---

## [Editor Report · Acceptance letter]

25 Oct 2024

PCOMPBIOL-D-24-00398R1 

Spatiotemporal orchestration of calcium-cAMP oscillations  on AKAP/AC nanodomains is governed by an incoherent feedforward loop

Dear Dr Rangamani,

I am pleased to inform you that your manuscript has been formally accepted for publication in PLOS Computational Biology. Your manuscript is now with our production department and you will be notified of the publication date in due course.

With kind regards,

Lilla Horvath
